# Hydrogel Use in Osteonecrosis of the Femoral Head

**DOI:** 10.3390/gels10080544

**Published:** 2024-08-22

**Authors:** Zeynep Bal, Nobuyuki Takakura

**Affiliations:** 1Laboratory of Signal Transduction, WPI Immunology Frontier Research Center (WPI-IFReC), Osaka University, 3-1 Yamada-oka, Suita 565-0871, Osaka, Japan; 2Department of Signal Transduction, Research Institute for Microbial Diseases (RIMD), Osaka University, 3-1 Yamada-oka, Suita 565-0871, Osaka, Japan

**Keywords:** osteonecrosis, osteonecrosis of the femoral head, vascularization, bone grafts, bone regeneration, hydrogels, biomaterials

## Abstract

Osteonecrosis of the femoral head (ONFH) is a vascular disease of unknown etiology and can be categorized mainly into two types: non-traumatic and traumatic ONFH. Thus, understanding osteogenic–angiogenic coupling is of prime importance in finding a solution for the treatment of ONFH. Hydrogels are biomaterials that are similar to the extracellular matrix (ECM). As they are able to mimic real tissue, they meet one of the most important rules in tissue engineering. In ONFH studies, hydrogels have recently become popular because of their ability to retain water and their adjustable properties, injectability, and mimicry of natural ECM. Because bone regeneration and graft materials are very broad areas of research and ONFH is a complex situation including bone and vascular systems, and there is no settled treatment strategy for ONFH worldwide, in this review paper, we followed a top-down approach by reviewing (1) bone and bone grafting, (2) hydrogels, (3) vascular systems, and (4) ONFH and hydrogel use in ONFH with studies in the literature which show promising results in limited clinical studies. The aim of this review paper is to provide the reader with general information on every aspect of ONFH and to focus on the hydrogel used in ONFH.

## 1. Introduction

As the second largest transplant tissue in the world, regeneration studies regarding bone are of prime importance. In bone grafting, although autografts are the gold standard, donor site morbidity, second operation, increased blood loss, and insufficient graft material are the disadvantages. As a second choice, allografts can be used; however, the lack of a proper allograft banking system and immune transmission probabilities brings the need for composite materials which can be designed for the desired properties depending on the purpose and site of application. Throughout these materials, hydrogels have recently become more and more popular because of their similarity to the extracellular matrix, which is a base for cells to settle on and is responsible for many crucial events such as cell-to-cell adherence, molecular signaling, cell migration, proliferation, and so on. Since hydrogels are also perfect materials for soft tissue regeneration, they are known to promote vascularization, which is also a prerequisite for the bone regeneration process. There is a fine balance between osteogenesis and angiogenesis. When this balance is disrupted, this can lead to many bone diseases, such as osteonecrosis of the femoral head (ONFH). 

ONFH is a vascular disease in which the blood supply to the femoral head is disrupted either by a traumatic or non-traumatic reason. Throughout the non-traumatic ONFH cases, steroid-induced ONFH is the most common type, followed by alcohol-induced ONFH, and then the others, such as hematological disease-related, genetic, or idiopathic ONFH. Total hip arthroplasty (THA) is the last resort for treatment in general. However, in the case of early necrosis, which in clinics is defined as the state just before collapse, generally as a minimally invasive surgery, core decompression surgery is chosen worldwide. But even after core decompression surgery, depending on (1) the level of necrosis, (2) the amount of necrotic tissue, (3) the other diseases that the patient has, or (4) the patient’s history as well as their age, the healing process may be delayed, and the femoral head may still collapse because of the distracted mechanical balance. Therefore, very recently, research on hydrogel use in ONFH has become popular among scientists and surgeons mainly because of their injectability and their adjustable properties based on the desired purposes. 

In this review, we first give brief information about bone, bone regeneration, and the graft types used in bone regeneration. Then, we give very brief general information about our focused case, ONFH. Then, we mention the use of hydrogels in bone regeneration studies with additional brief sections about commonly used polymers, ions, growth factors, and other substitutes in bone regeneration. Because angiogenesis is indispensable for osteogenesis and bone regeneration, following the hydrogel section, vascularization is explained briefly with a subsection on bone vascularization. In the last part, the studies in the literature regarding hydrogel use in ONFH are summarized. The literature research was conducted via PubMed and Google Scholar by using the keywords “hydrogel + osteonecrosis of the femoral head” and “hydrogel + femoral head necrosis”, and all accessible studies from any time interval until 2023 related to hydrogel use in osteonecrosis of the femoral head were included in this paper. 

With this review, we aim to show the general picture of how bone tissue engineering, vascularization, and hydrogels interplay in a specific disease case: ONFH. 

## 2. Bone and Bone Grafting

As the largest tissue in the body that is responsible for detoxification, maintaining the body shape, working as an endocrine organ, and providing the mineral balance of the body, and as the second largest transplant tissue in the world, regenerative approaches for bone tissue are one of the hot topics in science [1,2,3]. Natural bone can basically be divided into two parts: inorganic and organic. The organic part is mainly made up of collagen, and the inorganic part is mainly composed of nano-hydroxyapatite crystals [3,4,5,6]. In the United States alone, every year, 6.3 million people suffer from bone fractures, and 25% of bone fracture cases are reported to be in need of bone grafting [3,7]. However, the limitations related to bone grafting are challenging for regeneration studies.

Bone cells can basically be divided into four cell types: osteoblasts (OBs), osteocytes, bone-lining cells, which are differentiated from the osteoprogenitor lineage, and osteoclasts (OCs), which are derived from the hematopoietic cell lineage (Figure 1). Bone lining cells are “quiescent flattened osteoblasts” that elongate on the bone surface when the bone is neither in a resorptive nor in a formative phase [8,9]. By producing osteoprotegerin (OPG) and the receptor activator of nuclear factor kappa beta (NF-_K_B) ligand (RANKL), they play a role in osteoclast differentiation, and they are also reported to prevent osteoclast and bone matrix interaction under non-resorptive phase conditions [9]. Osteoblasts are cuboidal cells in their active form that are found on the bone surface where there is active bone formation and can be described basically as bone-forming cells that secrete bone extracellular matrix (ECM) proteins such as collagen type I (COLI), osteocalcin (OCN), osteopontin (OPN), and also alkaline phosphatase (ALP) and provide matrix mineralization [8,9,10]. As the most abundant bone cell type, osteocytes comprise 90–95% of the total bone cells [9]. They are basically the osteoblasts entrapped in the matrix they produce, located in the lacunae, and are known as bone cells [8,9]. Mohammed A.M. (2008) mentioned that these cells have a widespread distribution in bone, i.e., 25,000 cells per mm^3^ of bone, and this abundance makes them perfect sensors for mechanical stress on bone tissue. Therefore, they are excellent mechanoreceptors in bone tissue [8]. As the large multinucleated cells differentiate from the hematopoietic lineage, osteoclasts are known as bone-resorptive cells. They differentiate into mature osteoclasts (or, in other terms, terminally differentiated osteoclasts) mainly by macrophage colony-stimulating factor (M-CSF) secreted by osteoprogenitor mesenchymal cells and RANKL secreted by osteoblasts, osteocytes, and stromal cells [9,10].

During bone remodeling, the balance between osteoclasts and osteoblasts is of prime importance. If this balance favors osteoclasts over osteoblasts, a decrease in bone mineral density (BMD) will occur. As a result, the bones will become fragile and easy to fracture, with no or delayed healing, and fibrosis will occur after a fracture. If this system abruptly favors osteoblasts over osteoclasts, unwanted bone formation will occur such as ectopic bone formation/heterotrophic ossification. If the resorptive function of osteoclasts is restricted, for example, because of a mutation such as occurs in osteopetrosis (characterized by increased bone density), handicapped osteoclasts and disrupted bone resorption would negatively affect osteogenesis. In addition to the importance of the balance between bone formation and resorption, the balance between osteogenesis and angiogenesis, known as osteogenic–angiogenic coupling, is another hot topic in bone development and regeneration. As with the importance of oxygen, nutrients, growth factors (GFs), cytokines, hormone carriers, and the bone vasculature, so too is the balance between osteogenesis and angiogenesis of prime importance to bone regeneration and bone regeneration studies. Therefore, in the case of a bone-related disease, a bone graft should be designed carefully, in terms of both osteogenesis and angiogenesis, to provide proper bone regeneration and healing.

Bone grafting, which is indispensable for the treatment of defects larger than a critical size—a defect size that does not heal by itself spontaneously but needs other interventions for regeneration—can basically be divided into four groups: (1) autografts, (2) allografts, (3) xenografts, and (4) composites (Figure 2).

In many applications, autografts are generally seen as the gold standard because these are grafts from a patient’s own tissue and are advantageous because they pose a low or no risk of immune reactions and rejections. Additionally, autografts have intrinsic osteoconductivity and osteoinductivity, and are biodegradable. However, the need for a second operation, an insufficient amount of graft material, prolonged operation times, and donor site morbidity or complications, such as hematoma, infection, and chronic pain, are disadvantages [3,7,11,12,13,14]. In comparison, allografts are the second most popular graft type because they are slightly osteoconductive and osteoinductive, and most do not show complications such as donor site morbidity, excess blood loss, and the need for a second operation. However, disease transmission or immune rejection are disadvantages since allografts involve grafting from one human to other. For xenografts, the main problems are ethical and societal concerns because of the transplantation from one species to others. Porcine grafts are presently the most popular xenograft materials. For both allografts and xenografts, the other disadvantage is the lack of allograft/xenograft banks.

Therefore, the limitations of autografts, allografts, and xenografts bring the need for composite materials or scaffolds that can be designed as desired to use as grafts in BTE. The expected properties of an optimal graft include being osteoconductive and osteoinductive, nonimmunogenic, biocompatible, mostly biodegradable, and having the proper mechanical properties depending on the application site. The properties of a graft used in BTE is expected to have the following components: (1) scaffold; (2) GFs/cytokines/biochemical cues or substitutes; and (3) cells (mainly progenitor/stem cells) [2].

A scaffold is a material that acts as an anlage for cells and provides tissue development while also behaving as a mechanical support for newly regenerating tissue. In 1960, the first-generation materials that aimed to replace lost tissue with a minimal immune reaction were developed and included metals such as titanium and titanium alloys, ceramics such as alumina and zirconia, and synthetic polymers such as poly(methyl metacrylate) (PMMA) and polyetheretherketone (PEEK) [15,16]. For decades, metallic implants were used because of their load-bearing capabilities; however, they are not biodegradable and are likely to lead to pseudo-allergic hypersensitivity. Such implants also show pseudo-tumor potential, toxicity, and corrosion leading to the probability of implant failure that results in the need for a second operation in the future. They may also cause a change in the chemical environment, because of the ions released from the metallic implant via corrosion, affecting the cells and the surrounding tissues [3]. Additionally, these first-generation materials are mostly bioinert with a low or no integration ability with the surrounding tissues. Second-generation materials, with the advantages of having a bioactive nature and sometimes being biodegradable, are divided into synthetic and natural, calcium phosphates (CaPs), calcium carbonates, calcium sulfates, or bioactive glasses. Third-generation materials have the advantage of using materials, chemokines, or other substitutes with second-generation materials to induce specific cellular or tissue responses such as cell survival, differentiation, or specific lineage commitment. The aim of all such developments in materials is to find the perfect and desirable scaffold, which should have properties such as a suitable architecture for target tissues, and cyto- and tissue-compatibility. Depending on the target tissue, the scaffold should provide osseointegration and also be biodegradable and bioactive. Its mechanical properties should compensate for possible mechanical pressure or stress on the tissue.

The osseointegration ability of the graft or scaffold is crucial for the success of orthopedic surgery applications. Bioactive materials are significant for this ability because these materials allow the interaction of the material with the surrounding tissue upon transplantation and may contribute to the integration of the material with bone tissue [15,17]. Additionally, in BTE, the ability for osteoconduction and osteoinduction by the scaffold is primarily desired. Osteoinduction is the ability of materials to support new bone formation on the surface of the scaffold; in other words, the ability of materials to directly induce bone formation. Osteoconduction, which depends on the physicochemical characteristics of the material, is the ability of the material to absorb endogenous GFs, therefore allowing osteoprogenitor cells to migrate, differentiate, proliferate, and deposit on the material or within bone defects [2,5,15]. Also, osteoinductive materials can induce ectopic bone formation at the molecular, cellular, and tissue levels [3,15]. The materials’ surface characteristics, such as chemical composition, hydrophilicity/hydrophobicity, and topography, are also important factors that regulate cell behavior on material surfaces [15].

Growth factors are soluble secreted signaling polypeptides that can regulate cell differentiation and proliferation by binding to specific receptors on target cells. They are responsible for transmitting signals to regulate cytological and physiological processes. Thus, using GFs are one of the ideal methods for tissue regeneration. Growth factors are also able to act in small concentrations and are classified as local cytokines (which are proteins or peptides that allow the immune system fluids and the hematopoietic system to communicate). However, they have limited interactions with the ECM and are biologically unstable in varying heat and pH conditions, which causes them to have a poor stability and a short half-life as a result [18,19]. Also, the amount of GFs that are used should be adjusted properly. For example, in 2021, we assessed the regeneration capability of a novel composite of polylactic acid–polyethylene glycol/nHAp (PLA=PEG/nHAp) under different concentrations of recombinant human bone morphogenetic protein-2 (rhBMP-2) in a rat posterolateral fusion model (high dose, 10 μg and low dose, 3 μg) [13]. Although, in both groups, bony bridgings between L4–L5 transverse processes were achieved, micro-computed tomography (μCT; the higher the bone volume/tissue volume [BV/TV] value, the denser the bone) and hematoxylin–eosin (HE) and safranin O staining (SO) demonstrated that rats in the low-dose rhBMP-2 group showed denser bone formation than those in the high-dose rhBMP-2 group. Additionally, in the high-dose rhBMP-2 group, fatty bone marrow was observed in the newly formed bone tissue. From this, the idea arises that increased rhBMP-2 use can favor adipogenesis by activating the peroxisome proliferator-activated receptor gamma (PPARγ) pathway rather than osteogenesis.

Cells or substitute materials can also be added to scaffolds to initiate or support bone formation. However, the choice of the cells or substitutes to be used is of prime importance. For example, in the case of ONFH, the numbers of osteoblasts and osteocytes decrease and those of hematopoietic cells, as well as endothelial cells—the cells that line all blood vessels—in the bone marrow niche, decline in number. In this case, depending on the main aim, such as bone or vascular regeneration, a good choice is to use osteoprogenitor cells or vascular endothelial growth factor (VEGF)-overexpressing endothelial cells, respectively. Therefore, the cells or substitutes used should be chosen wisely depending on the target tissue, disease, or condition.

In tissue engineering, one of the main obstacles is ensuring the vascularization of the regenerative tissue [20]. It is also an important point for BTE and tissue/organ homeostasis because, most of the time, the success of the scaffold relies heavily on the vascularization characteristics of the construct [12,20,21,22]. In affecting bone regeneration or metabolism by discarding wastes, and providing oxygen, minerals, nutrients, and GFs to the target tissue as well as tissue-specific morphogenesis, vascularization plays a central role in BTE [2,22,23]. Thus, to promote the revascularization of tissue, the materials or scaffolds to be used can be designed to deliver proangiogenic GFs, nucleic acids, and/or vascular progenitor cells [24,25]. Im and Ling (2022) emphasized that depending on the target tissue, due to limitations in oxygenation and nutrient supply, avascularized constructs may show insufficient regeneration after in vivo transplantation [24].

For induction of the regenerative process or healing, neovascularization is mostly critical because newly formed vascular structures supply the target tissue with nutrients, GFs, stem and/or progenitor cells, and myeloid cells from the bone marrow or surrounding tissues to create a pro-regenerative niche [22,23,24,26,27,28,29,30]. Macrophages and neutrophils—as myeloid cells—are critical for graft vascularization because tissue resident macrophages are known to promote angiogenesis and mediate anastomosis between graft and host tissue. Meanwhile, non-inflammatory resident neutrophils secrete proangiogenic factors and modulate the inflammatory response as well as remodel the ECM to contribute to the vascularization of the graft material and the newly forming tissue [24,29,30].

Therefore, we can summarize the properties related to materials in BTE:To prevent any inflammation, including systemic inflammation, adverse effects, or material rejection in the target or surrounding tissues, the carrier scaffold should be non-toxic, non-immunogenic, and biocompatible.The material used in the scaffold should promote cell adhesion, proliferation, and mostly differentiation to promote target tissue–scaffold integration and tissue regeneration.Most of the time, depending on the purpose and transplantation site, the scaffold is expected to be biodegradable at a pace that does not harm the mechanical properties of the target tissue.

### Osteonecrosis of the Femoral Head

Although ONFH will be explained in detail later in this review, it is briefly introduced (Figure 3). Osteonecrosis of the femoral head can be defined as the vascular disruption of the proximal femur that leads to the death of osteocytes and bone marrow elements and concomitantly leads to bone necrosis [31]. Every year in the USA, 20,000 new cases of ONFH are reported. In the United Kingdom, the occurrence of ONFH is 1.4 per 100,000 which is almost similar to that in Japan at 1.9 in every 100,000 of the population [19].

In clinics, the treatment of ONFH is mainly based on the use of pharmacotherapy, surgical CD, porous tantalum rod implantation, osteotomy, and vascularized bone grafts [19,32]. However, more recently, hydrogels are being used after CD surgery to fill the defect since they have an ECM-like structure that is easily injectable into the defect site and sol–gel transition occurs in the body. There are also granular sponge bone grafts that can be used after CD surgery. However, after compaction into the drills, such grafts show osteoinduction only in the region of drilled tunnels/holes [33]. Thus, hydrogels with their ECM-like properties are a handy way to deliver cells, GFs, substitutes, or osteoinductive and osteoconductive materials to the target site without the concern of dissemination or, in other words, leakage and shape concerns. Transforming growth factor beta (TGFβ), BMPs, fibroblast growth factors (s), VEGF, and insulin-like growth factors (GFs) are mainly used as GFs in bone regeneration. However, problems with the use of these GFs are their short half-life, their side effects in cases of excessive use, and their leakage from the target tissue to the surrounding tissues or the whole body [7,13,32]. Hence, natural polymer-based hydrogels of alginate, chitosan, gelatin, collagen, and hyaluronic acid, or synthetic polymer-based hydrogels of polyvinyl alcohol (PVA), polyethylene glycol (PEG), polycaprolactone (PCL), and poly-lactic-co-glycolic acid (PLGA), or a combination of both—semi-synthetic polymers—can be used in combination with CD surgery in treating ONFH [2,26,34].

In tissue engineering, natural materials are generally chosen because of their hydrophilic surface properties that ease cell adherence, proliferation, and differentiation [5]. However, because natural polymers have low mechanical properties, high stability, and high batch variability, synthetic polymers that are adjustable and controllable with low or no batch variability are chosen over natural polymers. Yet, the disadvantages of synthetic materials include low biocompatibility and lower biological activity compared to natural polymers.

In regeneration studies, the best and most desirable scaffolds are the ones mimicking the natural tissue environment and derived either from natural or synthetic materials. Hydrogels can mimic the ECM that provides structural and signaling support to related tissues and cells for proliferation, migration, and cell adhesion. Hence, because of their soft and wet surface that increases the affinity for the ECM to enhance tissue–material integration, hydrogels are becoming popular materials in the area of bone regeneration and, more recently, ONFH.

## 3. Hydrogels

Extracellular matrices are known for their well-organized protein and polysaccharides network structures that not only act as three-dimensional (3D) supporting matrices and mechanical support for cells, but also act as a regulator in cell communication. For example, endothelial cells can connect to ECM via integrin receptors that can recognize many ECM proteins, such as laminin (Glossary Point 1), elastin (Glossary Point 2), fibronectin (Glossary Point 3), collagen, and additional soluble factors and cell surface proteins. Thus, ECMs also conduct and regulate the behavior of organs depending on physiological needs such as tissue regeneration [35,36,37,38,39,40,41,42,43].
Glossary Point 1:**Laminin** is one of the main components of the basement membrane and is interwoven with collagen type IV networks in the basement membrane by nidogen [35,41]. Laminins are effective in cell adhesion, migration, and differentiation, are synthesized by almost all epithelial cells, and are indispensable for early embryonic development and organogenesis [35,41]. Because integrins can recognize laminins, laminins mediate their cell–ECM interaction via integrins.

Glossary Point 2:As an insoluble polymeric protein, **elastin** is responsible for the elasticity and resilience of extracellular matrix and, therefore, can be defined as a natural polymer that gives elasticity to natural tissues. Yet such elastin stretch is limited by collagen–elastin association [40,41,42,43]. Elastin is a characteristic protein of soft tissues especially in blood vessels and the lungs

Glossary Point 3:Additionally, **fibronectin** (also known as biological glue) is a component of the basement membrane. At least 20 dif-ferent fibronectin molecules have been characterized and these are essential for cell adhesion and migration [41,44]. Fi-bronectin is activated by binding to a cell surface receptor and is then assembled into fibrils [44].

Additionally, proteins in the bone matrix are crucial for ensuring mechanical strength and tissue adhesive characteristics [44]. Based on protein proportion, structure, and location, the ECM can also be categorized as the “interstitial ECM,” mainly characterized by collagens I and II; and the “vascular basement membrane,” characterized by the presence of laminin and collagen type IV. These two different ECMs may induce different signal transduction pathways [6,39,41]. In the ECM, collagen is responsible for the tensile strength and elastin is important for the elastic recoil properties of the tissues [36,38,45,46]. Additionally, as a highly mineralized tissue, bone ECM can also be defined as a solid intercellular matrix of organic and inorganic phases [33].

Hydrogels are polymer networks that are widely used as carriers for biological materials in the context of BTE [47,48,49]. For a material to be called a hydrogel, it must have an absorption potential of at least 10–20% of its weight [47,49]. The absorption potential or solute diffusion are determined by its free versus bound water and the amount of water in the hydrogel [47].

An ideal composite or scaffold should be similar to natural tissue in order to provide proper regeneration. Hydrogels are mostly chosen for their similarity to ECM, which is the non-cellular component of tissues and organs. This is because in the body, most cells, apart from blood cells, are anchorage-dependent cells that need a solid matrix such as ECM [36,42,45,46,50,51]. Thus, hydrogels are ideal scaffold candidates. They are also an ideal scaffold because of their ability to mimic the characteristics of the extracellular matrix, such as the stimulation of cell migration, differentiation, and apoptosis. In addition, they can provide mechanical strength and behave like a message center for cells and tissues in signal transduction.

A hydrogel can also be defined as a 3D polymer network that is able to retain huge amounts of water due to its hydrophilic structure. This absorption amount is dependent on the crosslinker type and concentration, which also affects the swelling behavior and mechanical strength of the hydrogel [11,21,35,47,48,52,53]. For example, by adjusting the gelatin and glutaraldehyde concentrations of a gelatin hydrogel, the water content can be changed [35]; this plus collagen crosslinking are known to play critical roles in the stimulation of angiogenesis [2]. Also, the different hydration proportions of hydrogels can regulate cell adhesion behaviors [21].

The advantages of hydrogels are their biocompatibility, tunable biodegradability, low friction coefficients, controllable mechanical characteristics, porous structure, and osteoconductive and osteoinductive properties [2,24,33,39,48,51,54,55]. However, it is better to emphasize once more that the main advantage of hydrogels is their similarity to the ECM, which is one of the necessities in the bone regeneration process with progenitor cell homing, osteoid mineralization, osteoblast and osteocyte formation, and vascularization. Another important property that makes hydrogels preferable is the gelation that can take place under physiological conditions and that eliminates surgical interventions or additional surgeries [52]. Yet, the low mechanical strength and fragile nature of hydrogels are still negative properties that can be compensated for by combining hydrogels with calcium phosphate ceramics such as nano/micro HAp, β tricalcium phosphate (TCP), and biphasic calcium phosphates (BCP) [3,56].

Hydrogels can be divided into subclassifications based on their origin, polymeric composition, configuration, physical appearance, and electrical charge, among other properties (Figure 4). Many good reviews and research articles exist related to the chemistry and synthesis techniques of hydrogels for different tissues [5,11,23,33,47,48,52,53,57,58]. However, in this review paper, we focus on the biological perspective and use of hydrogels. Therefore, the characterization or synthesis of hydrogels are not discussed in this paper. Additionally, in the literature that focuses on in vivo or in vitro regeneration studies, hydrogels are mostly grouped based on their origin of material as either natural or synthetic (Figure 5).

### 3.1. Natural Polymer-Based Hydrogels

In the preparation of natural hydrogels, two main types of polymers exist: polysaccharides and fibrous polymers [47,57]. The advantages of hydrogels made up of natural polymers are as follows: [5] their intrinsic cell–material interaction mechanisms; [2] the possession of natural enzymatic biodegradation mechanisms; [3] and their good biocompatibility [23]. However, having the potential for an immunogenic response, uncontrollable biodegradation rate, low mechanical strength and stiffness, low/poor osteoinductivity and osteoconductivity, and batch variability are the disadvantages of natural polymers. However, nanoparticles such as nHAps can be combined with polymer materials to overcome poor osteoconductivity and low mechanical strength [3,4]. The natural polymers commonly used in bone regeneration are collagen, gelatin, fibrin, silk/silk fibroin (SF), hyaluronic acid, alginate, and chitosan [2,12,23,35,56,57].

The characteristics of some natural polymers are outlined below.

#### 3.1.1. Collagen

As the most abundant protein in mammals, almost 90% of organic bone matrix is composed of collagen. Of 28 types of collagens identified in vertebrates, COL1 is the most abundant collagen type in bone ECM and collagen IV is the main component of the basement membrane [8,36,41,45]. They are responsible for tensile strength, cell adhesion, chemotaxis, and migration. In other words, collagens are directly responsible for tissue development and structural integrity [36,45]. As the main organic component of natural bone, collagen is also the gold standard natural polymer in polymer bone grafting.

#### 3.1.2. Gelatin

Gelatin, the product of partial collagen hydrolysis, is a biodegradable nontoxic mixture of peptides and proteins, and can be used as a protective agent [17,43,48]. Due to its enzymatic digestion, it has a poor mechanical stability but a high degradation rate or solubility under physiological conditions. However, the degradation rate of gelatin is adjustable and can be increased using different strategies by chemical adjustments of NH_2_ and COOH chemical groups [43]. GelMA, a frequently used hydrogel, is also a methacrylated type of gelatin.

#### 3.1.3. Fibrin

Fibrin is a non-soluble protein derived from a soluble protein, fibrinogen, by the protease activity of thrombin. It is involved in blood clotting (coagulation) mechanisms. It is generally used alone or in combination of other materials because it is known to improve cell attachment, proliferation, and differentiation [43].

#### 3.1.4. Silk/Silk Fibroin

As a natural biopolymer, SF is produced from *Bombyx mori* silkworm cocoons. It is preferred in tissue engineering because of its non-toxic and biodegradable nature, lower inflammatory reaction, and favorable mechanical strength [59]. Additionally, SF contains arginine–glycine–aspartic acid recognition sites that increase cell–material interactions via integrins [59].

#### 3.1.5. Hyaluronic Acid

Hyaluronic acid (also known as hyaluronan; HA) is a glycosaminoglycan molecule present in the ECM of adult cartilage and bone. Cartilage cells (also known as chondrocytes) have CD44 and RHAMM surface receptors that play pivotal roles in different cell mechanisms; by HA binding to these receptors, HA plays a role in chondrocyte differentiation [43]. It has previously been mentioned that HA synthesis occurs in large amounts in early bone formation. Hyaluronic acid is of prime importance for the normal longitudinal growth of limbs because it was shown that in a mouse model lacking the HA-synthesizing enzyme, the long bones were shortened [60]. Therefore, HA is used in both bone and cartilage tissue engineering despite its low biomechanical stability. This can be overcome by producing composites with stronger polymers, by adding proper calcium phosphate ceramics or substitutes, and/or by chemically functionalizing HA.

#### 3.1.6. Alginate

Alginate is a natural polysaccharide polymer derived from brown algae and consists of glucuronic and mannuronic acids. It is not only mostly used in different injectable formulations for cartilage regeneration studies but is also considered a potential hydrogel drug system for CD surgery [34,43]. Although it has no or low immunogenicity and toxicity, its poor mechanical properties are a disadvantage of alginate hydrogels.

#### 3.1.7. Chitosan

As one of the most abundant polymers, chitosan is a cationic polysaccharide that is derived from the exoskeletons of crustaceans by the alkaline deacetylation of chitin. With suitable biocompatibility and biodegradability, chitosan-derived hydrogel was shown to be a good candidate for bone and cartilage engineering especially because of its antibacterial efficiency, decreased foreign body reaction, easy incorporation of GFs, and its ability to promote cellular proliferation [43,59].

#### 3.1.8. Heparin

Heparin is an anticoagulant that upregulates BMP-2 expression and osteogenic differentiation and also increases VEGF and FGF expression. Modifications in heparin may allow the sustained release of drugs and proteins by protecting them from denaturation and can improve the capacity of drug and protein loading [61].

### 3.2. Semi-Synthetic Hydrogels

Natural ECM components can also be modified by chemical modifications. This leads to another concept of polymers termed semi-synthetic materials. These are also called chemically modified natural biopolymers that are categorized under synthetic polymers, such as photocrosslinkable collagen–PEG hybrid material [2,26].

### 3.3. Synthetic Polymer-Based Hydrogels

Based on their longer degradation rate, adjustable mechanical strength, and high capacity for water absorption, synthetic hydrogels are superior to natural hydrogels [41,48]. They are mostly bioinert, which limits protein adsorption and so may prevent the undesired reactions of cell and protein attachments as well as the protein body response. Thus, these hydrogels may decrease possible hyperinflammatory reactions to foreign bodies [23]. Additionally, the use of synthetic hydrogels may allow the control of properties such as the degradation rate, toxicity, porosity, durability, mechanical strength, flexible structure, and batch variability [2,3,13,23,48]. Commonly used synthetic polymers are PLC, polylactic acid (PLA), polyethylene oxide (PEO), PVA, PEG, and PLGA. However, as synthetic materials, having a low biological activity is a negative property. These have been approved by the US Food and Drug Administration (FDA; Table 1) and are able to be modified, which generally compensates for their disadvantages.

The characteristics of some of these synthetic polymers are given below.

#### 3.3.1. Polyethylene Glycol

Polyethylene glycol is an FDA-approved photocrosslinked synthetic polymer characterized by hydrophilic properties. It is used in many bio-engineering approaches such as in modifying the degradation of biomaterials, decreasing viscosity, and lowering immunogenicity [34,43]. It is known to be resistant to protein absorption, which also makes PEG resistant to cell adhesion—a property that provides very low interference with incorporated bifunctionalities [66].

#### 3.3.2. Polylactic Acid

Polylactic acid is an FDA-approved synthetic polymer. It is an aliphatic biocompatible and biodegradable polyester with lactic acid byproducts that can be naturally metabolized [43]. It can be produced from poly-D-lactic acid (PDLA) or poly-L-lactic acid (PLLA), or from both [55].

#### 3.3.3. Polycaprolactone

Polycaprolactone has excellent biocompatibility and mechanical properties. It is approved by the FDA and so is clinically applicable. It is biodegradable. The degradation rate depends on the molecular weight, polymer crystallinity, and tissue environment [34].

### 3.4. Others

Thus, depending on the target tissue and desired therapeutic application, synthetic hydrogels can be constructed with cells, GFs, extracellular vesicles such as exosomes, cytokines, biochemical modifications, and substitutes (e.g., ions), among others [2,31,49,52,59,61,66,67].

Some of these are explained below.

#### 3.4.1. Exosomes

Exosomes are extracellular vesicles secreted by cells with diameters from 30 nm to 200 nm. They can enhance the effects of bone marrow stem/stromal cells (BMSCs) on immunomodulation and tissue regeneration [67]. Via modulation of the immune response, suppression of adipogenesis, and an improvement in osteogenesis, BMSC-derived exosomes can protect bone tissue from ONFH [67]. Exosomes have been shown to participate in vascular development, migration, and cell growth [68].

#### 3.4.2. Platelet-Rich Plasma

Platelet-rich plasma (PRP) is derived from peripheral blood containing a high concentration of platelets [65]. Having a high concentration of GFs, PRP can promote cartilage formation and osteogenesis [69,70] and can accelerate bone formation by promoting osteoblast proliferation.

### 3.5. Ions

Magnesium, silicon, and lithium are bioactive elements that may improve the osteoinductivity of bone cements; they are also used in hydrogels for the same purpose [59,67,71].

#### 3.5.1. Lithium

Lithium (Li) is a medically safe material shown to increase bone density by targeting the GSK-3β pathway to activate the Wnt signaling pathway, which is a key signal transduction pathway in bone regulation and regeneration [17,61].

#### 3.5.2. Magnesium

Magnesium (Mg) is a harmless osteoinductive metal that is abundant in the human body [3,12,59]. The cessation of bone growth, osteopenia, and decreased osteoblastic and osteoclastic activity are associated with Mg deficiency. Magnesium-based implants have been shown to promote bone healing and regeneration in vivo, probably because of their ability to promote integrin expression and the adhesion behaviors of osteoblasts [3,59]. Although it shows poor resistance to corrosion, Mg is emphasized to have an elastic modulus and comprehensive strength close to that of natural bone [3].

#### 3.5.3. Strontium

Strontium (Sr) cations are trace elements found in the human body that can promote bone formation. They have been shown to enhance BMD, bone remodeling, and mechanical strength even in osteoporotic conditions [3,70]. Strontium inhibits osteoclast formation by favoring the system for the OPG decoy receptor in OPG/RANKL signaling [3]. It can promote bone nodule formation, preosteoblastic cell proliferation, and the differentiation of progenitor cells to mature osteoblasts in a dose-dependent manner [3]. It should be noted that while low Sr concentrations may concomitantly decrease osteoclastic activity and bone resorption, increased Sr concentrations may inhibit calcium (Ca^2+^) resorption and may cause deleterious effects on bone mineralization [3].

### 3.6. Calcium Phosphates

Calcium phosphates (CaPs), such as HAps, TCPs, and BCPs, are a kind of bioactive inorganic ceramic similar to the main inorganic substitute of bone known as calcium phosphate apatite. They have a good biocompatibility and osteoconductivity which allows them to be considered as effective carriers for BTE [3,17]. Hydroxyapatite particles used together with hydrogels enhance mechanical strength while also contributing to the osteoinduction and osteoconduction of the hydrogel [3,4]. Also, citrates compromise 5 wt% of the organic components in bone and are known to affect bone development and load bearing function by binding to the surface of nHAp particles. They are thought to contribute to the stabilization of nHAp crystals [3,4]. Recently, bioactive octocalcium phosphates (OCP) are becoming popular in bone regeneration with hydrogels because of their similarity to HAps and ability to promote charge transfer between the material and tissue; they can be converted to HAps in the body because they are the precursors of HAps [72,73,74,75].

### 3.7. Growth Factors and Cytokines

#### 3.7.1. Transforming Growth Factor Beta 

Transforming growth factor beta (TGFβ) is one of the most important GFs and the most abundant cytokine in the bone matrix. It has roles in osteoblast and osteoclast activities, thus implicitly regulating bone homeostasis and bone remodeling [7]. Although TGFβ contributes to bone regeneration because of its osteoinduction activity, this is thought to be lower compared to that of BMPs [7]. Transforming growth factor beta is released to indirectly inhibit osteoclast activity and production by reducing RANKL secretion by osteoblasts and is known to recruit BMSCs during bone resorption.

#### 3.7.2. Bone Morphogenetic Proteins

Bone morphogenetic proteins (BMPs) are cytokines with roles in bone remodeling, based on their paracrine, autocrine, and endocrine characteristics, such as morphogenesis, fracture repair, cell proliferation, differentiation and migration, stem cell commitment, ECM remodeling, and apoptosis. Some BMPs are used in clinics in orthopedic applications because of their osteoinductive ability, especially for the healing of fractures and osteoporosis [7,49]. The FDA gave approval for the localized use of BMP-2 in anterior lumbar spinal fusion and tibial non-union fractures and BMP-7 for posterolateral spinal lumbar fusion and for complicated permanent tibial pseudoarthrosis treatments [7,49,56,76,77]. Bone morphogenetic protein-2 (BMP-2) is not only a potent stimulator of osteogenesis but also has effects on angiogenesis even under compromised bone healing conditions [7,78]. However, surplus amounts of BMP are related to ectopic and heterotrophic bone formation, carcinogenesis, the probability of metastasis shifting of adipogenic pathways rather than osteogenic pathways, and osteolysis [7,13,44]. Thus, while using BMPs, scientists and clinicians should be careful and should find proper scaffold materials to minimize the amount of BMP used.

#### 3.7.3. Fibroblast Growth Factors

Fibroblast growth factors (FGFs) have anabolic effects on bone formation and bone lineage cells, including chondrocytes, which are the main source of their secretion [27]. Fibroblast growth factor receptor 1 (FGFR1) and FGFR2, as FGF receptors, are also expressed in vascular cells [79]. Fibroblast growth factor receptor 2 can not only modulate VEGF expression on endothelial cells but can also increase FGFRs as well as VEGF receptors (VEGFRs) in endothelial cells [80]. Decreased trabecular bone was observed by others in mice lacking FGF. For example, in fracture healing, FGF9 is essential for the establishment of a vascular network. A decrease in the number of hypertrophic chondrocytes, the inhibition of chondrocyte proliferation, and delayed skeletal vascularization were shown to be related to the absence of FGF9 [79,80]. Additionally, increased bone vessel permeability and pericyte loss were observed after the endothelial cell-specific deletion of FGFR [79]. Therefore, FGF signaling regulates to both osteogenesis and angiogenesis [81].

#### 3.7.4. Platelet-Derived Growth Factor BB

As a GF, platelet-derived growth factor beta (PDGFβ or PDGFBB) stimulates many processes related to cell growth and differentiation. For example, PDGFBB secreted by murine osteoblasts increased bone strength and formation by promoting angiogenesis, osteogenesis, and nerve ingrowth [31]. Endothelial cell- and preosteoblast-derived PDGFBB play critical roles in the migration, proliferation, and differentiation of bone marrow-derived mesenchymal stem cells (BMMSCs) to promote angiogenesis and osteogenesis. Platelet-derived growth factor beta can also promote H-type vessel formation and bone regeneration by activating MAPK and PI3K/Akt signaling after binding to the PDGFBB receptor (PDGFR-BB) [79]. 

#### 3.7.5. Vascular Endothelial Growth Factor

Vascular endothelial growth factor (VEGF), which is a mitogen specific to endothelial cells, not only promotes angiogenesis but links osteogenesis and angiogenesis, thus having an important role in bone growth and repair [17,80,82,83,84]. A lack of VEGF in osteoprogenitors is related to a reduced bone mass and increased bone marrow fat in mice [82]. Vascular endothelial growth factor can promote new bone formation by stimulating revascularization of necrotic tissue and improving blood circulation, which can be taken as indicators of the strong angiogenic activity of VEGF [17]. Vascular endothelial growth factor can affect bone regeneration in two different ways [80]: In the first, it can directly act on endothelial cells to induce an angiogenic process that promotes the migration of progenitor cells to the bone callus; such cells differentiate into osteoblasts and thus osteogenesis occurs. The other pathway is through an angiocrine mechanism in which VEGF induces endothelial cells to produce osteogenic cytokines, such as BMP-2 and BMP-4, which directly differentiate osteoprogenitor cells into osteoblasts.

### 3.8. Main Properties of Hydrogels in Bone Regeneration

Characteristics such as the average pore size, distribution, and interconnection are also important for cell and protein attachments to hydrogels. Such pore properties also create an environment for bone cells to differentiate, proliferate, and migrate. In turn, this creates a strong interaction between hydrogel and bone tissue, and, concomitantly, a strong mechanical support for tissue [3,47,68].

Material porosity has a direct effect on cell-to-cell connection, migration, and proliferation; therefore, porosity is one of the important properties for proper tissue formation and function [3,7,85]. It is also important for the proper distribution of cells, oxygen, and nutrients throughout the material. Although high porosity and a large pore diameter are known to result in a higher diffusion rate and better diffusion of nutrients and oxygen, unfortunately, these properties also decrease the mechanical strength of the material. However, the mechanical strength may be increased by the addition of calcium phosphate ceramics [3,21]. For example, by adding nHAp particles to hydrogel, this increases the surface area which would promote cell attachment as well as the attached cell number. These would indirectly affect both the mechanical strength of the hydrogel and the newly formed tissue.

Liu et al. (2023) found that the pore sizes that favor nutrient transport and cell migration are around 100 μm, while vascularization and bone formation are supported by a pore size greater than 200 μm. While small pores of between 50 μm and 100 μm are optimal for endochondral ossification, large pore sizes of between 100 μm and 300 μm are shown to improve intramembranous ossification as well as vascularization [2,3]. Similar pore size–tissue related regeneration values are also reported by Annabi et al. (2010): the optimal pore size for neovascularization was found to be 5 μm, while from 100 μm to 350 μm was given as the optimal pore size range for bone regeneration, and from 40 μm to 100 μm for osteoid ingrowth [85].

The stiffness of hydrogels also affects cell viability, differentiation, migration, and host–graft cellular interactions [15,24,85]. Cells can respond to stiffness through mechanotransduction. In this manner, the change in stiffness can regulate cellular behavior and act on the determination of stem cell fate, such as if a stem cell differentiates toward an osteogenic or adipogenic lineage, among others. If the hydrogel is too stiff, cell viability may decrease resulting in limited or no spread and the migration of cells on/through the hydrogel. This may then likely cause host–graft failure due to a loss of cellular signaling 

The degradation rate and toxicity are other important characteristic of materials. If total tissue regeneration is the aim, biodegradable materials will be the best match since they do not have to be removed after transplantation. However, the byproducts of degraded materials should be non-toxic to prohibit any extreme immune reaction and material rejection [43,86]. The material degradation–stability balance or degradation rate is an important factor because, whether fast or slow, if the degradation rate is not in balance with the regeneration rate, unwanted complications could occur during the regeneration process [3,26,35,43]. The other important characteristics of any carrier scaffold, a “sustained release” ability—the ability of the scaffold to release the drug, GF, or cell at a desired rate in a certain period of time—is an indispensable property. In a way, it is also related to the degradation properties of the scaffold since fast degradation may result in the sudden release of GFs, ions, and cells from the hydrogel. This can interrupt sustained release, causing adverse effects in the surrounding tissue or at the metabolic level. For example, a sudden increase in the BMP-2 concentration in a transplantation environment may cause the infiltration of BMP-2 into the surrounding tissue and, concomitantly, ectopic bone formation [7,13]. As mentioned previously, many GFs also have a short half-life in the body. Thus, the fast release of GFs may interrupt their prolonged effects. Additionally, if the bone formation rate is slower than the degradation rate, a mechanical imbalance will occur, causing tissue collapse. Problems related to improper regeneration are also observed if the degradation rate of the hydrogel is too slow, which may cause retarded bone ingrowth within the defect. If the inflammation needed for the regeneration of the tissue is not resolved, the immune system will recognize the prolonged scaffold material as a foreign body. This will cause a fibrous encapsulation of the material, which would halt the proper integration of the surrounding tissue and material [86]. Therefore, the ideal is matched tissue regeneration—a material degradation rate with a suitable sustained release rate.

### 3.9. Vascularization in Bone Regeneration with Hydrogels

Hydrogels are also a favorite material in vascularization studies since they can be used as a temporal matrix because they have mechanical properties similar to those of soft tissues. They also mediate the delivery of regenerative therapeutic treatments as well as signaling between progenitor cells and pericytes [15,26,66]. Attaching proteins, proteoglycans, or inorganic substances, as well as cells and GFs, to hydrogels is another strategy to provide a feasible way to achieve desirable cell–material responses. For example, fibrin as a protein, heparan sulfate as a proteoglycan, and HAp as an inorganic substance are known to regulate vascularization via their high affinity for VEGF, epidermal growth factor (EGF), and basic (b)FGF [15].

However, in a clinical setting, the amount/size and stability of the hydrogel may be a problem depending on the size of the defect to be filled. This is because as the defect size increases, the problem of the perfusion of nutrients and oxygen through the hydrogel that fills the defect comes up. In biological tissues, oxygen can diffuse maximally up to only 100 μm–200 μm in depth [15,24]. To overcome this problem, in some studies chemical oxygen-generating biomaterials are used since the release rate of oxygen from oxygen-generating biomaterials plays a decisive role on the biological function of the material and target tissue [87]. However, the material should be designed properly because the burst release reaction of oxygen may lead to a sudden increase in oxygen in the tissue environment that would increase the reactive oxygen species (ROS) and the pH of the environment, causing a concomitant increase in toxicity. However, if the sustained release of oxygen from the materials is not available, extreme hypoxia will not support cell survival and this will lead to an incomplete vascularization of the target tissue; although a moderate degree of hypoxic environment is known to activate the hypoxia inducible factor 1 alpha (HIF1α)/VEGF pathway and promote angiogenesis [87]. However, the density of microvessels in the hydrogel construct or regenerative tissue itself can change depending on (1) the cell/gel metabolic demands, (2) the overall oxygen concentration, (3) the presence of pro-angiogenic/pro-osteogenic factors, (4) the cell type—if any—loaded on the hydrogel construct, (5) the transplantation site, and [7] the degradation rate or durability of the hydrogel [24,88].

Hydrogels aiming to provide vascularization should also be designed depending on the site of the defect or tissue and the needs of the tissue. Factors related to both the promotion and inhibition of vascularization can be used in hydrogels because insufficient vascularization may cause a lack of nutrients and oxygen at the target tissue and a lack of hydrogel–tissue integration; excessive vascularization is also a situation that is not always desired. Blache and Ehrbar (2018) discussed how hypervascularization may contribute to the pathogenesis of diseases such as cancer, psoriasis, and diabetic retinopathy [26]. 

Neovascularization is a fundamental part of tissue formation and repair. For hydrogels that are going to be used in BTE, compatibility with tissue and cells, being osteoinductive and conductive, are of prime importance. Considering the importance of vascularization in bone regeneration, designing hydrogels supporting angiogenesis additional to bone formation would also be an additional advantage of hydrogel.

## 4. Vascular System

The vascular system transports nutrients and oxygen to the tissues and organs, removes waste products and carbon dioxide from these, and plays critical roles in regulating osteogenesis and bone repair [2,22,83,89]. Endothelial cells that line the inner surface of vessels serve as an interface between blood and tissue. In development, there are basically two processes for forming blood vessels: (1) vasculogenesis, which is the *de novo* formation of blood vessels from endothelial precursor cells; and (2) angiogenesis, which is new blood vessel formation from pre-existing vessels by sprouting or intussusception (Figure 6).

Angioblasts, endothelial cell precursors derived from embryonic mesoderm, first differentiate into endothelial cells to form a “vascular plexus,” from which initial blood vessels are then formed by vasculogenesis; specification to arterial or venous endothelial cells is observed [25,81,83,90,91,92,93]. After this step, sprouting angiogenesis takes place for remodeling of the vasculature into a functional circulatory system [25,83]. Angiogenesis can be mostly divided into two parts: (1) intussusception which is the formation of new blood vessels by splitting vessels; and (2) sprouting angiogenesis, which is the formation of new vessels by endothelial cell sprouting. In sprouting angiogenesis, mainly four types of cells participate in: tip cells, breach cells, stalk cells, and phalanx cells. Basically, first breach cells breach the basal membrane to lead the way for tip cells during sprouting angiogenesis [89,94,95]. These former tip cells, also known as breach cells, are known to take roles in matrix remodeling and show increased expression of genes related to tip cells as well as collagen remodeling. Tip cells are cells that migrate along a VEGFA gradient by forming filopodia with the upregulation of delta-like canonical notch ligand 4 (DLL4) expression, which then activates notch signaling in neighboring cells and inhibits the tip cell phenotype leading to neighboring cells becoming stalk cells instead of tip cells [83,92] (Figure 7). Tip cells in the front anastomose to form a vascular network. In this process, vascular endothelial–cadherin (VE–Cad) expression and tissue macrophages are highly important because VE–Cad provides cell-to-cell junctions during anastomosis; tissue macrophages promote the bridging of tip cells [83,91,96]. Tip cell bridging and cell-to-cell junctions are followed by the further stabilization of new blood vessels via blood flow that creates shear stress that is then converted to biochemical signals by endothelial cells. Such biochemical signals related to shear stress by blood flow are regulated by signaling mechanisms, such as Krüppel-like factor 2 (KLF2) and Yes-associated protein (YAP). This leads endothelial cells to express PDGFBB and cause the recruitment of pericytes to further stabilize the vessels via the secretion of angiopoietin-1 (Ang-1). Such cells subsequently become phalanx cells, which are non-proliferative quiescent state cells that form a continuous monolayer. The intussusceptive angiogenesis mechanism is less clear. However, intussusceptive angiogenesis increases the surface area and volume of the vasculature in a relatively short time compared to sprouting angiogenesis. It should be noted that mature blood vessels are made up of different layer compositions. For example, capillaries have endothelial cells, pericytes, and a basement membrane; in larger blood vessels, the vascular walls are thicker and consist of mural cells and connective tissues that surround endothelial cells and the basement membrane [83].

As mentioned above, in endothelial cells, VEGF promotes angiogenesis. However, it can also regulate osteogenic GFs and stimulate osteogenesis. Thus, bone vasculature is an important element in the bone regeneration area.

### Bone Vasculature

Up to from 10% to 15% of cardiac output is received directly by bone. The bone vasculature creates a niche environment for not only bone cells but also for regulating the quiescence or mobilization of hematopoietic stem cells, which, together with progenitor cells, take positions very close to small arterioles and specialized sinusoids [82,84,97,98].

Not only endothelial cell types but also mesenchymal cells are essential for building blood vessels [24,84]. For example, endothelial colony-forming cells (ECFC), expressing platelet–endothelial cell adhesion molecule-1 (PECAM-1, also known as CD31) and VE–Cad, form lumens lining capillary networks. The presence of alpha smooth muscle actin-expressing (α-SMA^+^) mesenchymal stem/stromal cells (MSCs) is of prime importance because of their crucial role in the stabilization of ECFCs [24,83,84]. Additionally, the recruitment of myeloid cells, which are granulocytes and monocytes together, or myeloid lineage cells, such as macrophages, is also necessary for ECFC/MSC-mediated neovascularization [24,29,30]. For example, in both osteonecrosis and bone fractures, the bone vasculature is disrupted. This disruption leads to an inflammatory environment either by the death of environmental cells or by the formation of a hematoma. Under these conditions, macrophages are the first cells to arrive in the environment to resolve the inflammation. However, the M1/M2 macrophage ratio is one of the most important factors in tissue regeneration because, while M1-like macrophages are pro-inflammatory, M2-like macrophages are anti-inflammatory and known to promote vasculature formation.

Until 2023, bone tissue was known to have three endothelial subtypes that comprise L-type, H-type, and E-type blood vessels. In 2023, Iga et al. characterized a new type of endothelial cell subtype known as S-type (abbreviation for secondary ossification) endothelial cells in mouse long bone. These were solely observed in the epiphysis and secrete COLI, thus contributing to bone strength [97]. Of the remaining three subtypes of endothelial cells, E-type endothelial cells (CD31^hi^EMCN^lo^), abbreviated by Langen et al. (2017) because of their high abundancy in embryonic long bone at E16.5, are known to have a high capacity to support osterix (also known as transcription factor 7) positive/expressing (Osx^+^)-perivascular osteoprogenitors at the embryonic stage and in early postnatal bone. This subtype of endothelial cells, which is upstream of H-type endothelial cells, was observed to decrease in number during the postnatal period of life [40,98,99]. Such subtypes of endothelial cells are H-type and L-type endothelial cells and, with regard to H-type and L-type vascular structures, are the most studied endothelial subtypes in long bones. H-type endothelial cells, abbreviated because of their high expression of CD31 and endomucin (CD31^hi^EMCN^hi^), are known to be distributed mainly around the endosteum and metaphysis region and support trabecular bone repair, osteogenesis, and bone regeneration [40,84,98,100,101,102,103,104,105,106] (Figure 8).

Runt-related transcription factor-2 (RUNX2)-, the master transcription factor of osteogenesis COL1 alpha-, and Osx-expressing (RUNX2^+^Col1a^+^Osx^+^] osteoprogenitors, and platelet-derived growth factor receptor beta- and neuron-glia antigen 2 -expressing (PDGFRBB^+^NG2^+^) perivascular cells (Glossary Point 4), are specifically located around H-type capillaries and contribute greatly to osteogenesis. H-type vessels are highly distributed, especially around areas with high metabolic and osteogenic activity [98,101,102,104]. Additionally, Osx^+^-osteoblast precursors have been described as guiding vessel invasion, which can be taken as proof of osteogenic–angiogenic coupling [99,104,105,106]. Thus, VEGF, notch (Glossary Point 5), HIF1α, Slit guidance ligand 3 (SLIT3), and PDGFBB are some of the factors and mediators that regulate osteogenesis and H-type vessels, in other words, angiogenesis [2,40,80,84,98,100]. In bone remodeling, mechanical loading induces PDGFBB secretion by macrophages and non-resorbing osteoclast lineage cells and recruits endothelial and osteoblast precursor cells [106]. Also, preosteoclast-secreted PDGFBB stimulates osteogenesis by increasing the H-type vessel number in ovariectomized mice [99].
Glossary Point 4:**Perivascular cells** refer to both pericytes and smooth muscle cells; their precursors are embryonic fibroblasts. Perivascular cells work together with endothelial cells to form healthy blood vessel structures because they are responsible for the stabilization of blood vessels.

L-type endothelial cells (CD31^lo^EMCN^lo^), abbreviated according to their low expression of CD31 and EMCN, are downstream of H-type vessels. They form discontinuous, fenestrated, highly branched, and highly permeable bone marrow sinusoids at diaphysis [40,90,98,107]. Bone marrow leptin receptor-expressing (LepR^+^) cells and chimeric antigen receptor (CAR) cells are two types of perivascular cells that are associated with L-type vessels [81]. In contrast to H-type vessels, these sinusoidal cells have no relationship with Osx^+^ osteoprogenitors. As aging takes place, the number of H-type capillaries decreases while the L-type sinusoid numbers remain almost the same.
Glossary Point 5:**Notch signaling** is one of the important signaling pathways in cell–cell interactions and is positively regulated by blood flow. Its activation in soft tissues inhibits the proliferation and sprouting of endothelial cells and negatively regulates angiogenesis. In bone, the activation of notch signaling enhances angiogenesis and osteogenesis. The activation of endothelial cell notch signaling induces H-type vessel maturation and expansion [78,104,107]. Therefore, inhibition of the notch signaling pathway results in impaired bone formation. Notch signaling activation in endothelial cells is a prerequisite for their proliferation. It promotes the production of noggin, which is responsible for the proliferation and differentiation of perivascular cells, and also affects osteoprogenitors [107]. Noggin secreted by the angiocrine pathway is one of the modulators of the skeletal patterning of ossification [102].

These H-type capillaries, with lower permeability compared to L type sinusoids, are also observed to be located nearer to arterioles. They take nutrients and oxygen directly from these arterioles, which leads them to exhibit a higher partial O_2_ pressure. Therefore, the environment around H-type capillaries generally shows lower ROS levels [79]. Increased mechanical loading through increased body weight and muscle contraction causes H-type vessels to change to L-type vessels [99].

## 5. Osteonecrosis of the Femoral Head

Ischemic osteonecrosis (ON) is caused by the disruption of the blood supply to a region of the bone. Although the molecular mechanisms of ONFH have not been elucidated as yet, the absence or diminishment of blood flow to the related bone area causes an ischemic injury that leads to an inability to maintain cell viability. This process also causes the accumulation of metabolic waste products in the tissue environment that may also lead to the leaking of proteolytic enzymes into the surrounding tissues [24]. Thus, the revascularization of necrotic tissue is essential for bone healing in ON.

Osteonecrosis of the femoral head (also known as avascular necrosis of the femoral head) is a progressive disease in which the local destruction of osteocytes and bone marrow cells, due to hypoxia and ischemia, occurs as a result of the disruption of the blood supply to the bone and interrupted repair process. These develop into necrotic bone resorption and structural deformities [12,70,76,108]. Osteonecrosis of the femoral head mostly affects young adults aged between 30 and 50 years of age and is characterized by a high disability rate. It mostly occurs bilaterally, in more than half of cases [12,32,34,76,109]. The causes of ischemia in ONFH may include mechanical interruption, intravascular occlusion, extravascular compression, or all of these (Figure 9).

An imbalance between intravascular occlusion, fibrinolysis, and a coagulation system, such as in hypercoagulation, are other reasons for ONFH [12,109]. If left untreated, ONFH may cause subchondral bone collapse, pain, secondary arthritis, and hip and joint dysfunction [12,17,110].

Osteonecrosis of the femoral head can be mainly divided into two types based on etiology: (1) traumatic and (2) non-traumatic (Figure 10).

Traumatic ONFH is characterized by the mechanical disruption of blood vessels supplying the femoral head. Non-traumatic ONFH is more common in young and middle-aged adults due mostly to steroid and alcohol use, followed by hematological diseases such as sickle cell disease and polycythemia [12,110]. Traumatic osteonecrosis occurs in response to an acute mechanical disruption of the vascular structures in and around the femoral head. The occurrence of the initial lesions in the weight-bearing zone of the femoral head may cause a mechanical imbalance. In turn, this would lead to abnormal mechanical stimulation on the trabecular bone of the subchondral zone leading to concomitant stress fractures/microfractures [19,68]. Although trauma is also an inducible factor in non-traumatic osteonecrosis, the latter is often combined with other factors such as fat embolism, increased intraosseous pressure, and thrombosis, among other factors [68,110].

Steroids, also known as corticosteroids, are anti-inflammatory, anti-allergic, anti-toxin, immunosuppressive metabolic hormones with anti-shock potential. They are synthesized and secreted by the adrenal cortex and are the main reason for non-traumatic osteonecrosis of the femoral head. In bone tissue, steroids may lead to femoral head necrosis and osteoporosis involving matrix decomposition and the overexcretion of Ca^2+^ and phosphates and they may also cause the inhibition of osteoblastic activity and a reduction in protein mucopolysaccharides [68]. It was shown that long-term exposure to high levels of corticosteroids can disturb the osteogenic–adipogenic differentiation of BMSCs and may favor adipogenesis over osteogenesis by the inhibition of BMSCs and/or inducing their apoptosis [61,111] (Figure 11).

Steroid-induced ONFH is the most common ONFH type in world, making up almost half of all ONFH cases [17,34]. Luo et al. (2019) reported that in China, 26.35% and 55.75% of such cases are steroid-induced ONFH in men and women, respectively [17]. This is characterized by an impaired vascularization of blood vessels, decreased osteogenic activity or decreased differentiation of MSCs to osteogenic lineages, and chronic inflammation that hinders the normal bone repair process [67] (Figure 12). Inflammatory effects in osteoimmunology contribute greatly to steroid-induced ONFH and are mainly caused by macrophages since necrotic bone tissue stimulates macrophage inflammatory responses (Figure 13).

Macrophages, highly plastic cells, participate in tissue remodeling and inflammation and are essential in angiogenesis and vessel repair as well as bone regeneration. They can change their phenotype depending on the stimuli present in the tissue environment [112]. If stimuli are pro-inflammatory, the macrophage phenotype would mostly be M1-like. However, when the stimuli are anti-inflammatory, macrophages polarize to an M2-like phenotype [113]. M1-like macrophages, mainly relying on glycolysis, are responsible for producing ROS and nitrogen species to kill microbial pathogens, and they mainly produce inflammatory cytokines such as tumor necrosis factor alpha (TNFα), interleukin-1 beta (IL1β), IL6, IL12, IL18, and IL23 [112,113,114]. For the M2-like phenotype, these are alternatively activated macrophages that are mainly dependent on oxidative phosphorylation. They are mostly associated with angiogenesis, tissue growth, and morphogenesis, and the release anti-inflammatory cytokines such as IL10. The balance or ratio between M1-like and M2-like macrophages is important for ECM structure. For example, if M1-like macrophages polarize to anti-inflammatory, pro-reconstructive, pro-vascularization M2-like macrophages or if the balance favors M2-like macrophages, a more aligned matrix with thinner fibers is favored; a thinner fibrous layer is related to lower in vivo inflammation [28,67]. Additionally, increased M1-like phenotype macrophages are related to proinflammatory responses. Thus, an increase in the number of M1-like macrophages can cause chronic inflammation and contribute to steroid-induced ONFH. If the M1-like phenotype is shifted to an M2-like phenotype, this may create a favorable environment to stop the progression of steroid-induced ONFH (Figure 12).

Chronic alcoholism is another reason for ONFH since excessive drinking is shown to decrease bone cell metabolism, thus causing osteoporosis with thinner trabecular bone and making the bone structure more prone to local stress fractures [68] (Figure 14). Additionally, alcohol is shown to induce apoptosis in MSCs, which may cause necrotic areas in the femoral head leading to subchondral fractures [68,110]. It is also shown to induce lipid accumulation in MSCs and bone marrow, which increases the bone marrow cavity pressure. This, in turn, would apply pressure on inner blood vessels and block the circulation that evokes ONFH [110].

### 5.1. Pathology of Femoral Head Necrosis

Since it is difficult to collect samples from patients with ONFH in the clinic, characterizing very early pathological changes in ONFH is also difficult. However, in the literature, a necrotic femoral head is characterized as being a disrupted round structure of the femoral head with a yellowish opaque appearance and sparse discontinuous trabecular bone with cystic lesions [12]. In ONFH, the femoral head may exhibit an irregular contour, signs of collapse, and rough and pale cartilage (Figure 8).

The hematoxylin–eosin staining features of the bones in ONFH show a disordered bone structure with bone marrow necrosis, an increased number of empty osteocyte lacunae, and pyknotic nuclei [12,68,71,87,115]. The loss of nuclear staining in bone marrow cells and the increased number of empty osteocyte lacunae are well-known hallmarks of osteonecrosis [71,87,115,116,117]. Other features of osteonecrosis can be sparse or fragmented bone trabeculae with decreased osteocytes and bone marrow hematopoietic cells, adipocyte proliferation or hypertrophy, microvascular occlusion and/or reparative tissue designated by fibrosis, increased osteoclast activity, granulation tissue, multinuclear cell accumulation, and a delay in new bone formation [17,68,115,116,117].

The decrease in bone density and cystic degeneration of the femoral head can be observed in computed tomography (CT) or X-rays in the late stages of ONFH. The BV/TV value of the CT analysis indicates the bone density; in osteonecrosis, this value is mostly decreased. Also, the trabecular number (Tb.N) is the number of trabeculae in a defined bone area and is related to bone health. Decreased trabecular numbers are mostly observed in cases of osteonecrosis and secondary arthritis.

Acute inflammation that leads to native tissue restoration, fibrosis, and/or chronic inflammation is the first step toward healing in all tissue and organs. During the healing process of ONFH, multipotent progenitor cells and/or hematopoietic cells from the bone marrow are immobilized and differentiate into different cell types, such as vascular endothelial cells that contribute to neovascularization, osteoclasts that contribute to the resorption of necrotic bone, and osteoblasts that lead to subsequent bone formation [27,28]. Additionally, because of the microinflammatory environment, inflammatory cytokines and chemokines recruit innate immune cells, such as macrophages, neutrophils, and dendritic cells, and adaptive immune cells, such as B cells and T cells. These further release inflammatory factors that, in a positive feedback loop, increase the number of pro-inflammatory M1-like macrophages and decrease the number of neutrophils. In this manner, the overall inflammatory response is amplified [28,112,114,118]. It is emphasized that macrophages are locked into the pro-inflammatory M1-like macrophage phenotype in the case of osteonecrosis; they continuously release TNFα, IL1β, IL6, IL12, and IL18 and exacerbate injury [113,118]. An osteogenic response will be interrupted by persistently high levels of inflammation.

A diagnosis of ONFH mainly depends on magnetic resonance imaging (MRI), X-rays, and CT results. In T1-weighted MRI, ONFH presents as a low-intensity signal area while it shows a high-intensity signal area on T2-weighted MRI; MRI is the best tool for an early diagnosis of ONFH [32]. For X-rays and CT, crescent signs indicate the presence of ONFH. However, a crescent sign means collapse has already started. Thus, especially in X-rays, a crescent sign is not useful for an early diagnosis of ONFH. Additionally, although CT results are not superior to those of MRI, they are sensitive to detecting subchondral fractures and can be used as a supplementary diagnostic method.

Treatments for ONFH may basically be divided into two: non-operative and operative therapies [12,68]. In turn, non-operative therapies can be grouped into two: drug and physical therapies. However, around 80% of patients given conservative treatments will experience femoral head collapse within a few years; thus, the therapeutic effect of non-operative treatments does not seem very promising for now. Operative treatments are bone grafting, femoral head preserving surgeries, such as CD or osteotomy, and total hip arthroplasties (Figure 15).

Total hip arthroplasty (THA) is considered as the last resort for treatment of the hip joint, especially in cases of osteonecrosis and osteoarthritis. This surgical procedure can successfully reconstruct hip joint function. However, if the patient is too young, s/he should refrain from having this surgery because future operations may be required due to the wear and inadequate fixation of the prosthesis [12,68,119,120]. It is thought that by 2030, rates of THA will increase by 71%. Of all procedures billed to Medicare USA, THA is one of the most expensive [120]. Thus, this brings on the need for femoral head-preserving surgeries that may apply to ONFH in the early stages. These include CD, one of the most often chosen methods since it can mitigate the high intraosseous pressure of the femoral head by removing necrotic bone while stimulating bone cell growth and improving microcirculation in the area. However, decompression holes with larger diameters may cause problems of decreased mechanical strength, subchondral fractures, and iatrogenic collapse. It addition, it may not always induce osteogenesis or vascularization and thus brings on the need for the use of non-vascularized or vascularized grafts [12].

Although CD is one of the most popular minimally invasive surgical techniques used, it is mostly used in Europe and United States but not in Japan, except for biopsy purposes. Kuroda et al. (2016) reported the clinical collapse rate of the femoral head after CD surgery was more than 70%; meanwhile, it is reported to be 50% after rotational osteotomy [32]. However, because there is no standard surgical treatment for ONFH, CD is still the most applied minimally invasive surgery for the early stages of ONFH worldwide. In clinics, CD and bone grafting are the most common treatment strategies for ONFH [17].

After CD surgery, injecting regenerative materials/scaffolds into the empty canal is a good strategy to regenerate the necrotic head, increase mechanical support after surgery, and to mitigate the negative effects of necrosis. Therefore, hydrogels, because of their injectable nature and adjustable properties (if the disadvantages of their mechanical properties are omitted), are one of the most suitable materials to overcome the hurdle of regeneration after a CD operation. This is because they are injectable and can adopt different shapes in real time, their gelation may occur in situ, and leakage of the GFs can be prevented.

One of the ideal strategies for the treatment of ONFH is delivering effective GFs or cytokines to the necrotic bone by injecting hydrogels into the canal of CD. Because injectable hydrogels can be constructed under mild conditions and are similar to extracellular matrix, they are good materials to be used for minimally invasive surgery also known as keyhole surgery, especially after CD surgery [32,43,76,117]. They are also mostly viscous at room temperature and solidify at the site of injection (sol–gel transition), which makes them superior from the perspective of minimally invasive surgery [11].

### 5.2. Hydrogel Use in ONFH in the Literature

To date, the literature has recorded very few studies related to hydrogel use in the treatment of ONFH. Yet the numbers of articles are increasing because of the previously mentioned advantageous properties of hydrogels, especially for supporting minimally invasive surgeries (Table 2).

#### 5.2.1. Basic Research

##### Studies Including Data Related to Osteogenesis Only

In one such study, Fu et al. (2023) developed a novel, heat-sensitive, nanocomposite hydrogel system with a secondary structure. This was used to deliver gene fragments to regulate Bcl-2 and PPARᵧ gene expression in an alcohol-induced necrotic femoral head model in rats. The novel hydrogel was also assessed for sustained release and whether it could be applied in clinics [110]. A biguanide-modified-4-aminobenzoic acid (BGBA)-branched polyethylamine (bPEI) positively charged molecule was designed that absorbed negatively charged plasmids and small interfering (si)RNAs and self-assembled to form nanoparticles. These yielded genes encapsulated in nanoparticles that were than combined with a PLGA-PEG-PLGA hydrogel composite. Bcl-2 inhibits apoptosis in the stem cells of patients with ONFH. Peroxisome proliferator-activated receptor gamma is one of the main key regulators in adipogenesis. This novel heat-sensitive nanocomposite hydrogel provides the sustained release of nanoparticles that ameliorate alcohol-induced ONFH. It promotes bone reconstruction by inhibiting PPARᵧ expression by siRNA and by inhibiting stem cell apoptosis by inducing Bcl-2 expression using circular RNA-3503 (Figure 16).

In another study by Xu et al. (2021), an injectable, thermosensitive PLGA hydrogel loaded with long intergenic non-protein coding RNA 473 (LINC00473)-overexpressing rat-derived (r)BMMSCs was investigated in a steroid-induced ONFH rat model [104]. *LINC00473* is one of the differentially expressed genes from the bone marrow stem cells of steroid-induced ONFH patients compared to those in patients with a femoral head fracture as a control [111]. *LINC00473* was shown to increase ALP staining and expression of the osteogenic markers bone sialoprotein II (BSPII), osteopontin 3 (OPN3), and RUNX2 in human-derived bone marrow stem cells (hBMSCs). LINC00473 was shown to decrease the lipid droplet number in hBMSCs by triggering the Wnt/β-catenin pathway and also decreased adipogenic stimulation by inhibiting the expression of adipogenesis-related gene expression, such as peroxisome proliferator-activated receptor gamma (PPAR*γ*), CCAAT-enhancer-binding protein alpha (CEBPα), and fatty acid-binding protein 4 (FABP4). A decrease in triglyceride staining by oil red O was also reported. LINC00473 was also shown to rescue hBMSCs from dextran-induced apoptosis. Rat-derived bone marrow mesenchymal stromal cells were also shown to expand and adhere on a hydrogel scaffold as well as through pore walls (Figure 17). In vivo experiments also showed that hydrogels loaded with LINC00473-overexpressing rBMMSCs eased bone marrow edema, promoted osteogenesis in steroid-induced osteonecrosis, and inhibited adipogenesis. In both the studies mentioned, the authors were focused on the interplay between adipogenesis and osteogenesis. However, the importance of vascularization, or in other words angiogenic–osteogenic coupling, is of prime importance for ONFH. It would be better to check the effect of this composite on vascularization.

In a study performed by Ma et al. in 2023, gelatin–heparin–thymine (GHT) hydrogel was used as an BMP-2 carrier in an ischemic necrosis-induced piglet model of Legg–Calve–Perthes Disease. This is a juvenile form of ONFH, characterized by hip pain and a limited range of motion that may lead to physical disability [76]. The hydrogel was applied using multiple epiphyseal drilling surgeries after inducing ischemic necrosis by disturbing blood flow around the neck. Bone morphogenetic protein-2 was released in a sustained fashion for 4 weeks with a 10% burst release within the first 24 h. After a week, three epiphyseal drillings were performed, and hydrogels were injected following a saline wash. The hydrogel–BMP-2 treatment was shown to induce endochondral ossification in the subchondral region while not showing any signs of heterotrophic ossification. This meant that no leakage of the BMP-2 occurred despite the harsh osteonecrotic environment. In another study published in 2010, Kuroda et al. investigated the potential anabolic effects of 100 μg of human recombinant FGF-2 (rhFGF-2) via a single local injection of gelatin + rhFGF-2 gelatin hydrogel microspheres into the femoral head of adult Japanese rabbits in a steroid-induced ONFH model with electrocoagulation for the vascular occlusion of the capital femoral epiphysis [27]. The sustained release of rhFGF-2 from hydrogel microspheres was shown to continue for at least 2 weeks. Sixteen weeks after the rhFGF-2+hydrogel injection, in μCT results, an apparent regeneration of the trabecular bones and no femoral head collapse were noted while the progression of secondary osteoarthritis and osteonecrosis was inhibited in the rhFGF-2 treatment group. In the control group (hydrogel microspheres + phosphate buffered saline [PBS]), trabecular bone absorption, segmental collapse, and no trabecular bone regeneration were observed by μCT. In cadaver pig femoral heads, Phipps et al. (2016) used different concentrations of the peptide-based self-assembling hydrogel, RADA16 (PuraMatrix, 3D Matirc, Inc.), as a BMP-2 carrier vehicle. The backflow and distribution of this hydrogel was assessed since the uncontrolled leakage and spread of hydrogels represent a major problem [78]. They also checked the release of BMP-2, the gelation of different RADA16 concentrations with BMP-2, and the viability and proliferation status of pig (p)BMSCs, as well as the regeneration ability of RADA16 + BMP-2 hydrogels. A slight leakage of BMP-2 was observed that the authors thought was because of the lack of an inner circulatory system in the cadaver femoral heads. They observed that in the presence of BMP-2, high concentrations of RADA16 could not complete the gelation process that was related to high protein interactions. They observed that the BMP-2 released from RADA16 was able to induce the phosphorylation of SMAD1/5/8 signaling, showing that the BMP-2 released was able to induce the osteogenic pathway. They also found that the lower the concentration of RADA16, the greater the number of BMSCs. They emphasized the following studies related to the testing of RADA16/BMP-2 use in osteonecrosis.

In the three studies mentioned above, cytokine release from hydrogels and their effects with hydrogels on ONFH were observed. However, rhBMP-2/BMP-2 is known to facilitate neovascularization in a paracrine way and rhFGF-2/FGF-2 is known to have a role in the proliferation, migration, and differentiation of not only the osteogenic lineage but also of vascular endothelial cells. As a result, we think that it would be better if the effects of these hydrogels on vascularization were investigated at the same time. This is because, as mentioned above numerous times, ONFH is the necrosis of bone tissue due to compromised blood flow, which makes it a type of vascular disease. Additionally, in the last study mentioned here, although the experimental set-up was limited to in vitro and ex vivo models, it would have also been better to check the effects of BMP-2 release from the RADA16 hydrogel on vascularization.

##### Studies Including Data Osteogenesis and Neovascularization/Angiogenesis

Wang et al. in 2021 developed novel CaO_2_/gelatin microspheres composed of 3D-printed PCL/nHAp porous scaffolds. They tested the regeneration ability of this novel hydrogel scaffold in a steroid-induced ONFH rat model after CD surgery [87]. They focused on the O_2_ release rate as the most critical factor for the scaffold, composed of chemical O2-generating biomaterials, because this is the regulative value for the biocompatibility and biological function of the material. This novel composite was made up of three different subunits: (1) gelatin/CaCO_2_ microspheres that are the key component for sustainable O_2_ release; (2) a 3D-printed porous tube of PCL/nHAp for mechanical strength compensation of the scaffold; and (3) an alginate/gelatin hydrogel to provide a 3D environment for cell survival by filling the pores in the scaffold. They reported that the promotion of BMSC survival in vitro alleviated apoptosis and increased the expression of CD31^hi^, which is mostly expressed in regenerative H-type vessels, and thus increased bone reconstruction in steroid-induced ONFH samples.

In a study of an alcohol-induced ONFH model in SD rats by Yuan et al. (2022), after the ONFH model was created, 3 mm-diameter, 5 mm-depth defects were made from the femoral neck through the femoral head and the regenerative effects of hydroxypropyl-β-cyclodextrin–gelatin hydrogels (HPβCD/Gel), with or without BMSCs, were examined [121]. In the HPβCD/Gel + BMSC treatment group, an increase in osteocyte and chondrocyte number was observed; in addition, the BV/TV value was higher compared to HPβCD/Gel treatment alone. The new blood vessel area and density were also measured in both treatment groups. Although in the HPβCD/Gel alone treatment group, a new blood vessel area was observed, the new blood vessel density were greater in the HPβCD/Gel + BMSC treatment group. From this, it was concluded that angiogenesis in the HPβCD/Gel had direct effects on BMSCs. Additionally, the regenerated tissue was observed to be wider in area in the HPβCD/Gel + BMSC compared to HPβCD/Gel alone treatment group, which can be interpreted as the promotion of osteogenesis even in an alcohol-induced ONFH situation.

Another study, in which exosomes from Li-stimulated BMSCs (Li-exo) and conventional culture medium (Con-exo) were combined with methacryloylated type I hydrogel (lightgel group) to examine the effects of Li-stimulated, BMSC-derived exosomes on glucocorticosteroid-induced ONFH of SD rats, was performed by Chen et al. in 2023 [67]. First, exosomes were shown to be released for at least 2 weeks from the hydrogel. Then, the angiogenesis capacity of these three study groups was examined on human umbilical vein endothelial cells (HUVECs). It was found that VEGF expression was higher in the Li-exo/Lightgel and Con-exo/Lightgel treatment groups than in the Lightgel treatment group itself; in particular, the Li-exo/Lightgel group showed the highest VEGF expression. The more intense ALP and RUNX2 staining of BMSCs in the Li-Exo/Lightgel and Con-Exi/Lightgel treatment groups indicated the superior osteogenic ability of the exosome-including groups. In particular, the Li-exo/Lightgel treatment group was found to show better osteogenic differentiation of BMSCs. In vivo, after creating a glucocorticosteroid ONFH model in SD rats, CD surgery was performed and 2 mm-diameter, 3 mm-depth drilled holes were filled with hydrogel in the study and control (PBS) groups. A thinner fibrous layer, which is an indication of lower in vivo inflammation, was observed for the Con-Exo/Lightgel and Li-Exo/Lightgel treatment groups with the thinnest layer observed for the Li-Exo/Lightgel group. The Li-Exo/Lightgel group exhibited the highest expression of arginase 1, which is an anti-inflammatory regenerative M2-like macrophage marker, and lower inducible nitric oxide synthase (iNOS), which is a pro-inflammatory M1-like macrophage marker, followed by the Con-Exo/Lightgel, Lightgel, and control groups. The authors concluded that Li-Exo/Lightgel promoted osteogenesis and that these gels supported the M2-like macrophage phenotype. Thus, Li-engineered BMSC exosomes can promote and accelerate the therapeutic bone healing process in glucocorticosteroid-induced ONFH. The authors measured the vascularization capability of the hydrogels on HUVECs since ONFH is a disorder directly related to vascular damage and the authors were aware of the importance of vascular regeneration. However, in our opinion, vascular structures or vessel densities should have also been determined in vivo.

**Table 2 gels-10-00544-t002:** Summary of research papers related to hydrogel use in ONFH.

Ref	Material	Experimental Model	Cell/Model Animal	Results
[110]	siRNA and plasmid-absorbed BGBA-bPEI molecules + PLGA-PEG-PLGA heat-sensitive nanocomposite hydrogel system	Alcohol-induced ONFH model Intrafemoral head hydrogel injection	In vitro: MSCs In vivo: Rats	Secondary nanostructure of the hydrogel achieved sustained release of siRNA and plasmids. In vitro: On MSCs, PPARᵧ was inhibited, and Bcl-2 expression was increased. In vivo: Empty osteocyte lacunae and apoptosis were lower in the hydrogel-treated group. Osteogenic activity was increased as well as trabeculae. The hydrogel was successful in promoting ONFH-associated lesions.
[111]	Injectable thermosensitive PLGA hydrogel loaded with LINC00473-overexpressing rat-derived bone marrow mesenchymal stem cells (rBMMSCs)	Steroid-induced ONFH	In vitro: hBMSCs, rBMMSCs In vivo: SD male Rats	In vitro: LINC00473 increased ALP staining and expression of the osteogenic markers BSPII, OPN3, and RUNX2 in hBMSCs, while it decreased the lipid droplet number in hBMSCs by triggering the Wnt/β-catenin pathway. It also decreased adipogenic stimulation by inhibiting expression of adipogenesis-related gene expression by PPARᵧ, CEBPα, and FABP4. rBMMSCs expanded and adhered on hydrogel scaffolds as well as through pore walls. In vivo: Hydrogels loaded with LINC00473-overexpressing rBMMSCs eased bone marrow edema, promoted osteogenesis, and inhibited adipogenesis.
[75]	GHT (gelatin–heparin–thymine) hydrogel + BMP-2	Ischemic osteonecrosis model of Legg–Calve–Perthes Disease (LCPD) induced by disturbing blood flow around the femoral neck Hydrogel injection by multiple epiphyseal drilling	In vitro: pBMMCs In vivo: Piglets	Sustained release of BMP-2 from the hydrogel was achieved for more than 4 weeks. In vitro: BMP-2 released from hydrogel was shown to preserve its bioactivity. In vivo: The number of empty osteocyte lacunae number was decreased in the hydrogel + BMP-2 groups, and epiphyseal bone regeneration and remodeling were accelerated. Additionally, necrotic bone was replaced by new bone in the hydrogel + BMP-2 group.
[26]	Gelatin hydrogel + 100 μg rhFGF-2	Steroid-induced ONFH + vascular occlusion A single local injection of hydrogel + rhFGF-2 into 1 mm-diameter, 5 mm-depth hole in the femoral head	In vivo: Adult male Japanese white rabbits	Apparent trabecular bone regeneration was reported. Progression of osteonecrosis and secondary arthritis was inhibited. No femoral head collapse was observed.
[77]	RADA16 (a peptide-based hydrogel) (PuraMatrix, 3D Matrix, Inc.)	Piglet model of ischemic osteonecrosis (cadaver)	In vitro: pBMSCs Ex vivo: Cadaver pigs	In vitro: Lower concentrations of RADA16 were better for cell survival and number. BMP-2 released from the surface of RADA16 was able to stimulate osteogenesis via the SMAD 1/5/8 pathway. Ex vivo: RADA16 was distributed properly through ischemic femoral heads with a slight backflow that might be related to not having an inner vasculature structure in the cadaveric femoral heads.
[86]	CaO_2_/gelatin microspheres composed with 3D-printed PCL/nanoHAp (nHAp) porous scaffolds (an oxygen- generating scaffold)	Steroid-induced ONFH Core decompression surgery + composite scaffold implantation	In vitro: Rabbit BMSCs In vivo: NZ white rabbits	In vitro: Experiments and the proliferation of BMSCs showed that the hydrogel composite scaffolds were able to generate oxygen in a sustainable manner. Co-culture experiments also showed the good biocompatibility and low toxicity of the scaffolds. In vivo: Exogenously transplanted BMSCs survived, the scaffold was able to mitigate cell apoptosis, and increased bone reconstruction was observed. Also, the alleviation of apoptosis and increased expression of CD31^hi^ were also reported. The composite can mitigate ischemic conditions by the sustained release of O_2_ and may provide bone and vascular regeneration under osteonecrotic conditions.
[121]	HPβCD/gelatin hydrogel with or without BMSCs	Alcohol-induced ONFH Core decompression surgery + hydrogel injection	In vitro: BMSCs In vivo: Male SD rats	In vitro: HPβCD/gelatin hydrogel was shown to be nontoxic HPβCD/gelatin + BMSC hydrogel showed a higher expression of the bone-related genes *ALP*, *OCN,* and *OPN* than HPβCD/gelatin hydrogel without BMSCs In vivo: HPβCD/gelatin hydrogel + BMSC group showed more trabecular bone compared to the HPβCD/gelatin hydrogel group without BMSCs. No collapse occurred of the femoral head in the HPβCD/gelatin hydrogel + BMSC group. For the HPβCD/gelatin hydrogel group, in some samples a regional collapse of the femoral head was observed. The HPβCD/gelatin hydrogel + BMSC group showed fewer empty osteocyte lacunae compared to the HPβCD/gelatin hydrogel group. The hydrogel was a good carrier of BMSCs and was shown to promote bone formation in the necrotic femoral head.
[66]	Exosomes from (1) Lithium-stimulated BMSCs (Li-exo) (2) From conditioned medium-stimulated BMSCs (Con-exo) + methacryloylated type I hydrogel	Glucocorticosteroid-induced ONFH Core decompression surgery + injection of hydrogel groups as Li-Exo/Lightgel, Con-Exo/Lightgel, Lightgel, and PBS	In vitro: BMSCs and HUVECs In vivo: SD Rats	Exosomes were shown to be released for at least 2 weeks from the hydrogel. In vitro: In HUVEC cells, VEGF expression was higher in the Li-exo/Lightgel compared to the Con-exo/Lightgel group. In BMSCs, more intense staining of ALP and RUNX2 was observed in the Li-Exo/Lightgel compared to the Con-Exo/Lightgel group. The highest expression of the anti-inflammatory and regenerative macrophage M2-like phenotype markers Arg-1 and CD206 and the lowest expression of the pro-inflammatory macrophage M1-like phenotype marker iNOS were observed in the Li-Exo/Lightgel group. In vivo: Coverage of the defect by collagen and new tissue formation were clearly observed in both the Li-Exo/Lightgel and Co-Exo/Lightgel experimental groups. The highest expression of BMP-2 was observed in the Li-Exo/Lightgel group. These results can be interpreted as meaning Li-Exo/Lightgel provides M1-like macrophage polarization to M2-like macrophages and promotes both osteogenesis and angiogenesis.
[116]	Injectable hydrogel (HG), based on collagen and alginate, delivered four different groups of rabbit bone marrow-derived MSCs (BMMSCs): (1) IL-4-overexpressing rabbit bone marrow-derived MSCs (IL4-MSCs) in normal medium (2) Rabbit bone marrow-derived MSCs (MSCs) in normal medium (3) Proinflammatory cytokine TNFα pre-conditioned medium + MSCs (pMSCs) (4) Pro-inflammatory cytokine TNFα pre-conditioned medium + IL4-MSCs (IL4-pMSCs)	Glucocorticosteroid-induced ONFH Core decompression surgery + HG injections	In vitro: IL4-MSCs, MSCs, IL4-pMSCs, pMSCs In vivo: Male NZ white rabbits	In vitro: HG-encapsulated MSCs were able to survive in the hydrogel for more than 21 days. Also, TNFα was shown to accelerate osteogenic differentiation. In vivo: pMSCs + HG were shown to support angiogenesis and increased BMD in the femoral head. IL4-MSCs accelerated proliferation and decreased the proportion of empty osteocyte lacunae.
[117]	VEGF-loaded temperature-sensitive star-shaped PLGA-mPEG block copolymer microsphere hydrogels composed of vascular endothelial cells	Alcohol-induced ONFH Injection of the hydrogel with VEGF and vECs to necrotic sites using a bone needle	In vitro: Vascular endothelial cells (vECs) In vivo: Japanese white rabbits	The release of VEGF from composites occurred in a sustained manner for 30 days. In vitro: Microsphere/hydrogels showed no negative effects on vEC viability. In vivo: The sustained release of VEGF was shown to provide a proper environment for vEC survival. Vascularization and bone formation was observed in the necrotic site. These results indicated that these microsphere/hydrogels, loaded with VEGF and composed of vECs, may be a good candidate for vascularization and osteogenesis.
[58]	rBMSC encapsulated, rhBMP-2 immobilized, Mg-loaded, chitosan/silk fibroid hydrogels	Femoral neck canal defect model and injection of hydrogel groups	In vitro: rBMSCs In vivo: SD rats	In vitro: ALP activity was upregulated and the mRNA expression of BMP-2, TGF-β1, RUNX2, COLI and OCN was increased. In vivo: The occurrence of vascularization was also observed, as well as osteogenesis. Bones and other organs showed no signs of toxicity. The hydrogels were also able to provide the sustained release of rhBMP-2.
[108]	HA-BP/CaP	Liquid nitrogen-induced ONFH via K-wire	In vitro: MC3T3-E1 cells In vivo: Female NZ white rabbits	In vitro: The osteogenic differentiation markers ALP, OCN, VEGF, and COLI were increased in the hydrogel group. In vivo: The hydrogel group compared to the untreated saline control group showed an increased amount of regenerated collagen. In the hydrogel group, the trabeculae were not as sparse as seen in the control group. The number of empty osteocyte lacunae was also decreased in the hydrogel group.
[69]	PLGA/Sr/HAp hydrogels loaded with different sPL concentrations	Steroid-induced ONFH CD surgery + hydrogel injection	In vivo: Male SD rats	Sustained release of VEGF and TGFβ from the hydrogel was achieved over a 30-day period. In vivo: The hydrogel + sPL was shown to trigger osteogenesis and vascularization.
[30]	PGK/PDGFBB/MSCs + HG	Steroid-induced ONFH CD surgery + HG injection	In vitro: MSCs In vivo: Male NZ white rabbits	In vitro: The PGK group showed a higher proportion of ALP and alizarin red staining than the CMV group, although no significant difference was noted for alizarin red staining between the groups. In vivo: The number of empty osteocyte lacunae was decreased and angiogenesis increased in the PGK/PDGFBB-MSC+HG groups.
[60]	Heparin lithium hydrogel (Li-hep-gel) + miR335-5p-pendant tetrahedron DNA nanostructures (niR@TDN)	Steroid-associated ONFH CD surgery followed by hydrogel injection	In vitro: BMSCs In vivo: Rabbits	In vitro: Many vascular-like structures were observed in miR@TDN+li-hep-gel. VEGF expression was also higher in miR@TDN+li-hep-gel. In vivo: Li-hep-gel reduced the number of empty osteocyte lacunae. The lowest number of osteocyte lacunae was observed in the miR@TDN+Li-hep-gel group. VEGF staining was highest in both the Li-hep-gel and miR@TDN+li-hep-gel groups. β-catenin staining was highest in the miR@TDN+li-hep-gel group. New tissue formed in the miR@TDN+li-hep-gel group was more compact compared to that in the Li-hep-gel group.
[109]	Gelatin hydrogel + 800 μg rhFGF-2	A single local shot of gelatin hydrogel + rhFGF-2 to ONFH patients	One-year clinical follow-up (n = 10)	In total, 9 out of 10 patients did not show any sign of femoral head collapse. An increased bone mass in deficit areas of the patients was also observed.
[120]	Gelatin hydrogel + 800 μg rhFGF-2	A single local shot of gelatin hydrogel + rhFGF-2 to ONFH patients *A multicenter Phase II trial clinical study performed in 4 hospitals in Japan (The University of Tokyo, Gifu University, Osaka University, and Kyoto University)*	Two-year outcomes (n = 64)	Joint preservation time was reported to be increased at the two-year follow-up (≥65%).

siRNA, small interfering RNA; BGBA-bPEI, biguanide-modified-4-aminobenzoic acid (BGBA)-modified branched polyethylenimine (bPEI); PLGA, also known as PLA, poly(lactic-co-glycolic) acid; PEG, polyethylene glycol, also known as polyethylene oxide (PEO) or poly(oxyethylene) (POE) depending on molecular weight; ONFH, osteonecrosis of the femoral head; MSCs, mesenchymal stem/stromal cells; PPAR*γ*, also known as PPARG, peroxisome proliferator-activated receptor gamma; Bcl-2, B-cell leukemia/lymphoma protein-2; LINC00473, long intergenic non-protein coding RNA 473; rBMMSCs, rat bone marrow mesenchymal stem/stromal cells; ALP, alkaline phosphatase; BSP II, bone sialoprotein II; OPN3, osteopontin 3; RUNX2, Runt-related transcription factor 2; CEBPα, CCAAT-enhancer-binding protein alpha; FABP4, fatty acid-binding protein 4; pBMMCs, porcine bone marrow mesenchymal cells; LCPD, Legg–Calve–Perthes disease; BMP2, bone morphogenetic protein 2; rhFGF-2, human recombinant basic fibroblast growth factor 2; RADA16, an ionic self-complementary peptide RADARADARADARADA; CaO2, calcium peroxide; PCL, polycaprolactone; nHAp, nano-hydroxyapatite; HAp, hydroxyapatite; BMSC, bone marrow stem/stromal cells; NZ white rabbits, New Zealand white rabbits; HPβCD, 2-Hydoxypropyl-β-cyclodextrin; OCN, osteocalci;. OPN, osteopontin; HUVECs, human umbilical vein endothelial cells; SD Rats, Sprague Dawley rats; VEGF, vascular endothelial growth factor; iNOS, inducible nitric oxide synthase; IL4, interleukin 4; TNF-α, tumor necrosis factor alpha; BMD, bone mineral density; vECs, vascular endothelial cells; Mg, magnesium; TGFβ-1, transforming growth factor beta 1; COL1, collagen 1; HA-BP/CaP, bisphosphonate (BP)-modified hyaluronic acid (HA) and calcium phosphate (CaP); MC3T3-E1, C57BL/6 mouse calvaria cell line characterized by increased ALP activity; CD surgery, core decompression surgery; Sr, strontium; sPL, super activated platelet lysate; PGK/PDGFBB/MSCs, platelet derived growth factor beta (PDGFBB) gene carrying lentivirus vector transduced rabbit mesenchymal stem cells (MSCs) under the control of phosphoglycerate (PGK); CMV, cytomegalovirus; HG, collagen–alginate hydrogel.

In 2021, Maruyama et al. published a study of a corticosteroid-associated ONFH model in rabbits in which they used an injectable hydrogel (HG) based on collagen and alginate to deliver four different groups of rabbit-derived BMMSCs to the femoral head after CD surgery at a 3 mm diameter and 2 mm depth [116]. In this study, the effect of the pro-inflammatory cytokine, IL4, was tested since IL4 polymorphism may be related to steroid-induced ONFH. The BMMSC groups were (1) IL4-overexpressing rabbit bone marrow-derived MSCs (IL4-MSCs) in normal medium; (2) rabbit bone marrow-derived MSCs (MSCs) in normal medium; (3) proinflammatory cytokine TNFα pre-conditioned medium + MSCs (pMSCs); and (4) pro-inflammatory cytokine TNFα pre-conditioned medium + IL4-MSCs (IL4-pMSCs). Exposure to the pro-inflammatory cytokine TNFα accelerated the osteogenic differentiation observed in vitro. In vivo, pMSCs + HG were shown to support angiogenesis and increased the BMD in the femoral head. IL4-MSCs accelerated proliferation and decreased the proportion of empty osteocyte lacunae.

Chen et al. (2018) developed injectable temperature-sensitive star-shaped PLGA-mPEG block copolymer microspheres by loading VEGF into a linear PLGA-mPEG block copolymer to form hydrogels and then added composed vascular endothelial cells (vECs) to these hydrogels [117]. They tested the hydrogel’s ability to regenerate bone and vascularization in an alcohol-induced ONFH model in Japanese white rabbits. They showed the sustained release of VEGF from the composites for 30 days; these microspheres/hydrogels did not have any negative effects on vEC viability. Also, in vivo experiments showed the sustained release of VEGF from the microsphere/hydrogel composites created an excellent environment for vascular endothelial cell survival that induced vascularization and osteogenesis concurrently.

In 2021, Lu et al. designed a novel hydrogel of rBMMSCs, encapsulated rhBMP-2, and immobilized Mg incorporated in chitosan–SF, and assessed its ability to deliver rBMMSCs into a femoral head necrosis site by injection in a femoral neck canal defect model of SD rats [59]. In vitro, it was shown that ALP activity was upregulated and that the mRNA expression of the bone-specific extracellular proteins BMP-2, TGB-β1, RUNX2, COLI, and OCN was increased. Also, in vivo*,* the occurrence of vascularization was observed as well as osteogenesis. Bones and other organs, such as the liver, heart, spleen, kidneys, and lungs, were harvested to investigate the toxicity of the hydrogels; no signs of toxicity were observed. The hydrogels were able to provide the sustained release of rhBMP-2.

In 2019, Wang et al. demonstrated the positive osteogenic effects of an injectable hybrid hydrogel of bisphosphonate modified hyaluronic acid + calcium phosphate (HA-BA/CaP) on liquid nitrogen-induced ONFH in female NZ white rabbits [108]. The hydrogel was able to increase the number of Ca^2+^ nodules and ALP staining in mouse embryonic osteogenic precursor cells (MC3T3-E1) as well as the expression of genes related to osteogenesis, such as *ALP*, *OCN*, *COLI*, and *VEGF*. The hydrogels were also shown to increase the number of osteoblasts and collagen I fibers and decrease the number of empty osteocyte lacunae in vivo. Although the authors demonstrated increased VEGF expression in vitro, in our opinion the angiogenic effect of the hydrogel in vivo should also have been determined.

In 2021, Huang et al. developed a novel hydrogel of temperature-sensitive PLGA + SrCL2 + HAp hydrogels loaded with super active platelet lysate (sPL) (PLGA/Sr/Hap + sPL) and tested these in a steroid-induced ONFH rat model [70]. The implantation of the hydrogels with different sPL concentrations was shown not to trigger any inflammatory reactions in vivo. The hydrogel + sPL was shown to trigger osteogenesis, as indicated by ALP and COLI tissue staining, as well as vascularization according to the CD31 staining of tissues.

Guzman et al. (2021) genetically modified MSCs to overexpress PDGFBB and used these with collagen + alginate HG in order to observe the effects in a rabbit model of steroid-induced ONFH [31]. In vitro*,* the human phosphoglycerate kinase (PGK) and cytomegalovirus (CMV) promoters were used to overexpress PDGFBB. The PGK group showed a higher proportion of ALP and alizarin red staining than the CMV group, although the alizarin red staining was not significantly different between the groups. In vivo, the number of empty osteocyte lacunae was decreased in the PGK/PDGFBB-MSC+HG groups; angiogenesis was also reported to be increased.

In 2022, Li et al. developed a hydrogel of heparin lithium (Li-hep-gel) + miR335-5p-pendant tetrahedron DNA nanostructures (miR@TDN) and tested the carrier abilities of Li-hep-gel for miR@TDN in a rabbit model of steroid-associated ONFH [61]. In vitro vascular-like structures were strongly observed in the miR@TDN+Li-hep-gel treatment group; VEGF expression was also higher in this group. In vivo, samples from rabbits in the Li-hep-gel treatment group showed a reduced number of empty osteocyte lacunae compared to the hep-gel group; the lowest number of empty osteocyte lacunae was observed in animals of the miR@TDN+Li-hep-gel treatment group. Also, VEGF staining observed to be highest in samples from animals in both the Li-hep-gel and n miR@TDN+Li-hep-gel treatment groups. Staining for β-catenin was shown to be the highest in samples from rabbits in the miR@TDN+Li-hep-gel treatment group. The new tissue formed in rabbits of the miR@TDN+Li-hep-gel group was shown to be more compact compared to animals in the Li-hep-gel group. However, in the latter group, trabecular bone formation was also observed and was found to be greater compared to animals of the hep-gel group.

#### 5.2.2. Clinical Studies

In a clinical one-year follow-up study by Kuroda et al. published in 2016, 800 μg of rhFGF-2-impregnated gelatin hydrogel was administered as a single local injection to ONFH patients (n = 10) at pre-collapse Stage 2 or lower. Only one patient developed femoral head collapse, while the other nine patients did not show any sign of collapse. An increase in bone mass in deficit areas of the patients was observed [109].

In another multicenter Phase II trial clinical study performed in four hospitals in Japan (The University of Tokyo, Gifu University, Osaka University, and Kyoto University) with 64 patients, 2-year outcomes of a single dose of 800 μg rhFGF + gelatin hydrogel treatment for ONFH were tested to see if rhFGF-2 was able to prevent femoral head collapse [122]. The joint preservation time was increased, and the joint preservation rate was ≥65%, which showed an increased clinical efficacy, radiological bone regeneration, and safety.

As mentioned previously, rhFGF-2 is directly related to vascular endothelial cell migration, proliferation, and differentiation and is a potent angiogenic factor. Thus, in our opinion, although these clinical follow-up studies show promising results for ONFH treatment, the change in vascular density should also have been measured.

## 6. Conclusions

As a vascular disease of bone, ONFH affects mostly 30- to 50-year-olds and the mechanism of disease has not been clearly elucidated as yet. Also, because it is generally difficult to define in the early stages, minimally invasive surgeries can only sometimes save patients before THA. Although CD surgery supported with biomaterial injections yields promising results, there is an ongoing need to design a proper injectable hydrogel composite. Also, because osteogenesis and angiogenesis cannot be separated from each other in the case of bone regeneration, when designing a hydrogel composite one should understand the following: (1) the type of ONFH; (2) the vascular structures in the site; and (3) the interplay between osteogenesis and angiogenesis in order to design an optimal hydrogel.

Although not many basic research or clinical studies exist regarding hydrogel use in ONFH as yet, the ECM-like properties of hydrogels makes these a popular focal point for further investigations in the future.

However, depending on our knowledge and the results of basic and clinical studies in the literature, and since ONFH is a vascular disease, focusing on vascularization is an indispensable route for the treatment of ONFH. We recommend scientists working in this area aim to better understand the relationship between osteogenesis and angiogenesis. They also need to avoid underestimating the interplay between bone cells and endothelial cells and hematopoietic cell interplay in the treatment of not only ONFH, but also other bone diseases, since these cells ensure homeostasis within the same niche.

To date, no review paper has focused on hydrogel use specifically for ONFH. Thus, as the first review paper focusing on this subject, we hope this review will be inspiring for further ONFH research.

## Figures and Tables

**Figure 1 gels-10-00544-f001:**
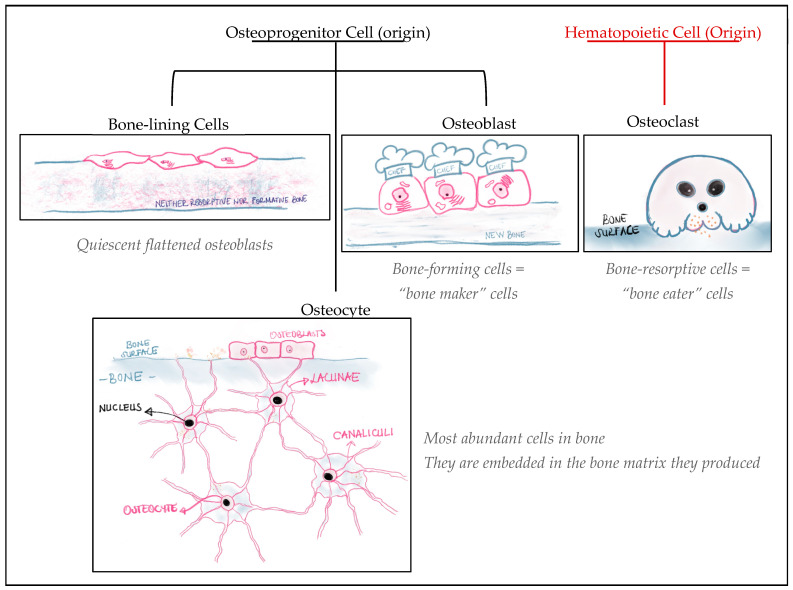
Bone cell types. Bone-lining cells, osteoblasts, and osteocytes are from an osteoprogenitor cell-line origin, while osteoclasts are from cells of hematopoietic origin. Bone-lining cells are quiescent flattened osteoblasts that likely prevent probable interactions of the bone matrix with osteoclasts in the non-resorptive phase. As bone-forming cells, osteoblasts are “the chefs of bone cuisine” because they make the bone. Osteocytes are the most abundant cells in bone tissue, embedded in the matrix they produce. They also act as mechanoreceptors. Osteoclasts are bone-resorptive cells that can be imagined as “bone eaters” that chefs, also known as osteoblasts, make. The balance between osteoblasts and osteoclasts is of prime importance for bone homeostasis. Any shift in the balance favoring either osteoclasts or osteoblasts may result in bone disease.

**Figure 2 gels-10-00544-f002:**
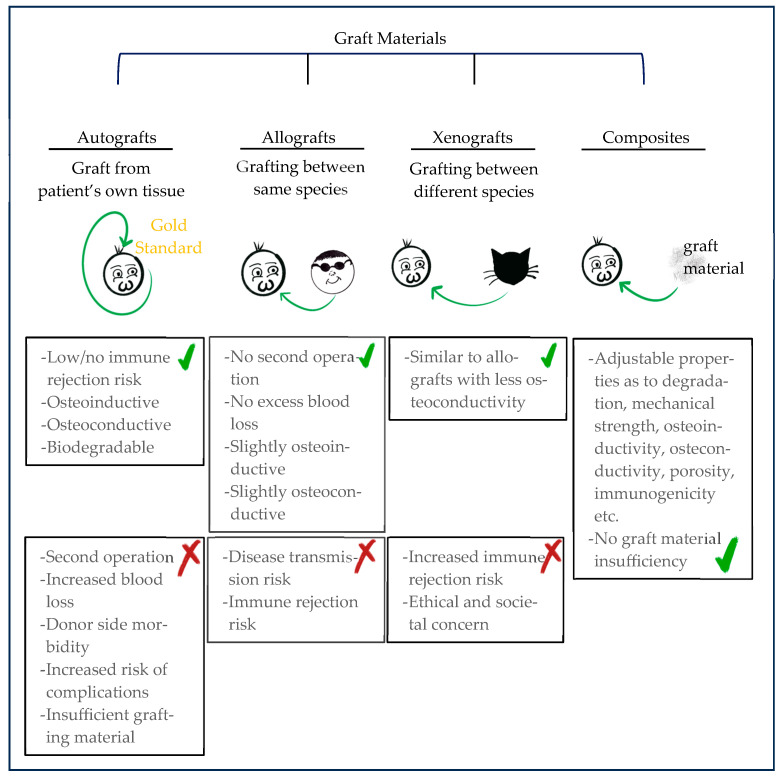
Classification of graft materials. The green check mark indicates the advantages of the related material while the red cross mark indicates the disadvantages.

**Figure 3 gels-10-00544-f003:**
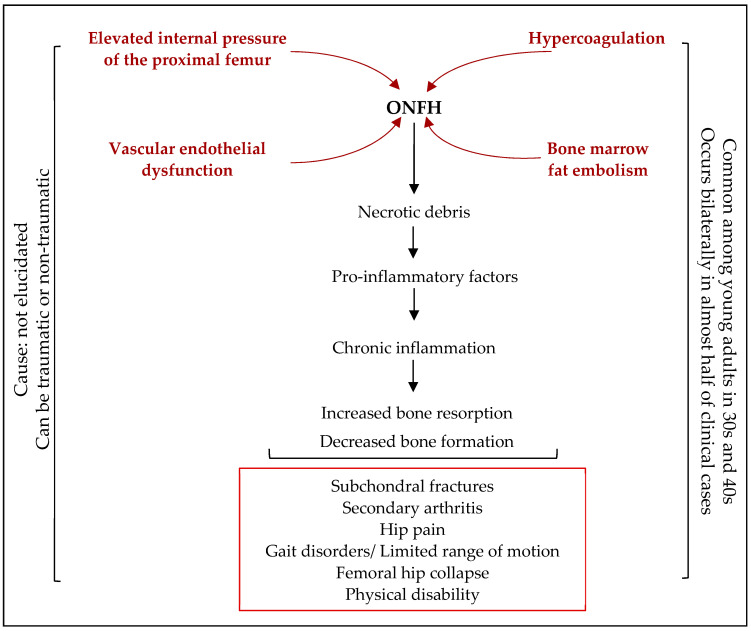
Brief summary of ONFH. Osteonecrosis of the femoral head (ONFH) is commonly seen in young adults in their 30s or 40s and generally occurs bilaterally in almost 50% of clinical cases. It may be traumatic or non-traumatic. Although the main cause of ONFH has not yet been elucidated, hypercoagulation, elevated bone marrow pressure of the proximal femur, vascular dysfunction, and bone marrow fat embolisms are viewed as contributing factors. The necrotic debris in ONFH releases pro-inflammatory factors that lead to persistent chronic inflammation resulting in concomitant increased bone resorption, disruption of the balance between bone anabolism and bone catabolism, and decreased bone formation. It can further lead to subchondral fractures and secondary arthritis, accompanied by hip pain, gait disorders, and limited motion. This highlights the progression of the disease to femoral hip collapse, which would lead to physical disability.

**Figure 4 gels-10-00544-f004:**
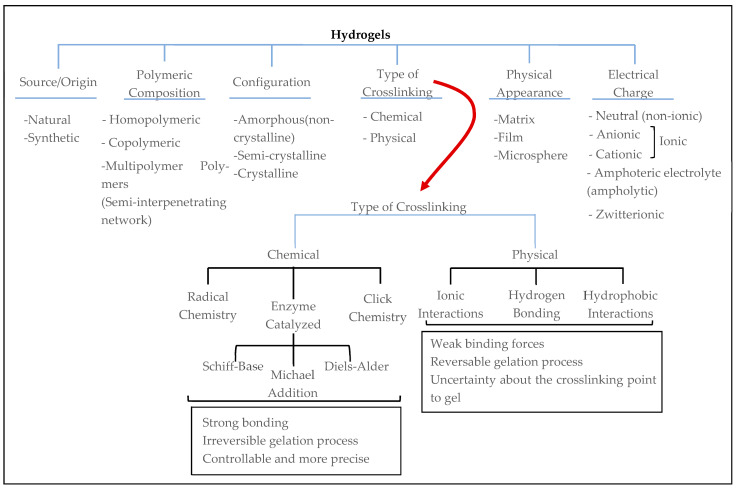
Hydrogel classifications. The type of crosslinking is given in detail to prevent any confusion for the reader since it is widely mentioned in the literature.

**Figure 5 gels-10-00544-f005:**
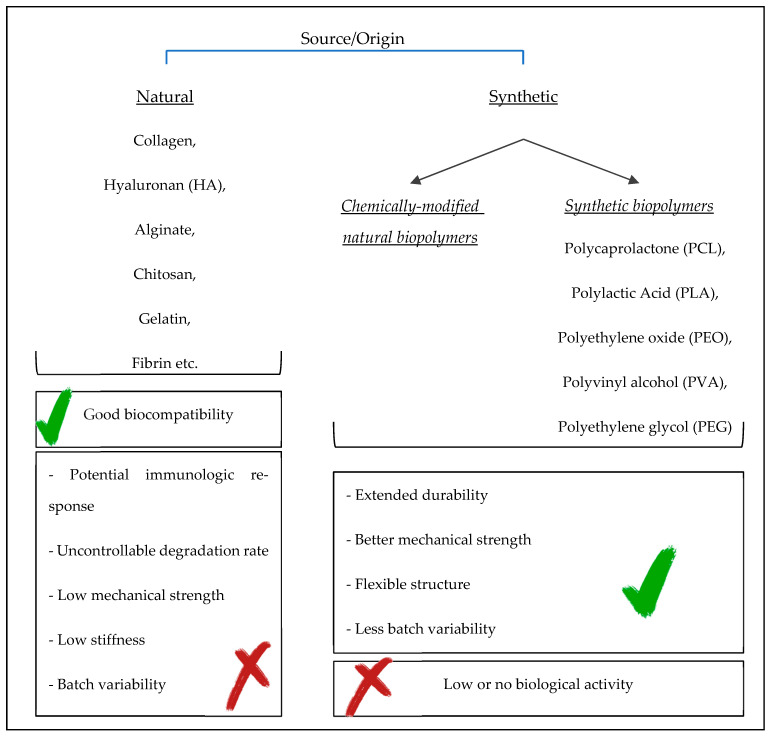
Hydrogel classification based on the source/origin of the hydrogel material. This paper is focused on this classification. Additionally, in the literature, chemically modified natural biopolymers are also known as semi-synthetic polymers and are grouped under synthetic polymers generally.

**Figure 6 gels-10-00544-f006:**
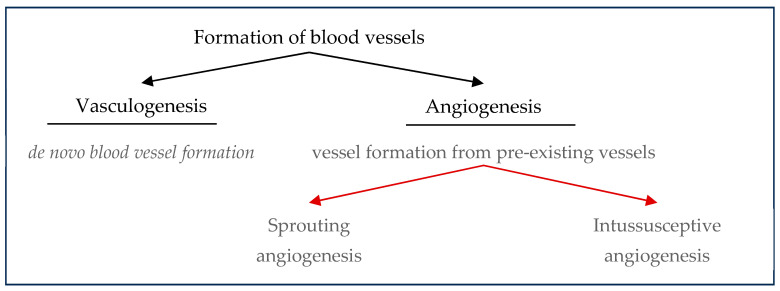
Scheme summarizing the processes of blood vessel formation.

**Figure 7 gels-10-00544-f007:**
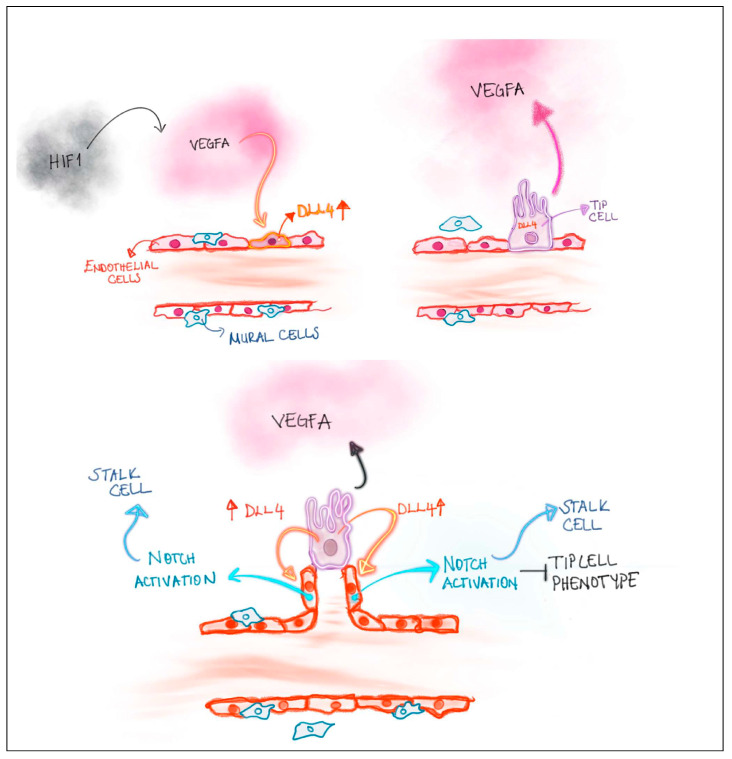
Sprouting angiogenesis. An increase in hypoxia-inducible factor 1 (HIF1) mediates the transcriptional activation of vascular endothelial cell growth factor (VEGF)A and increased VEGFA expression induces DLL4 expression in tip cells. Tip cells migrate along the VEGFA gradient and, as DLL4 expression increases, this causes notch activation in neighboring cells. Notch activation inhibits the tip cell phenotype in neighboring cells and induces a stalk cell phenotype. It is important to note that such phenotypes are interchangeable between cells.

**Figure 8 gels-10-00544-f008:**
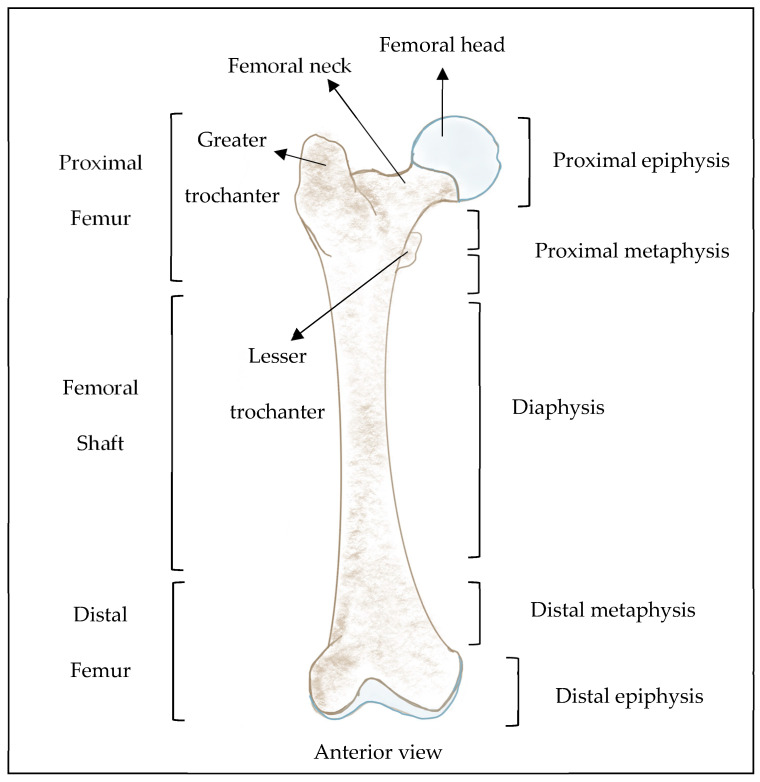
Parts of the femoral bone.

**Figure 9 gels-10-00544-f009:**
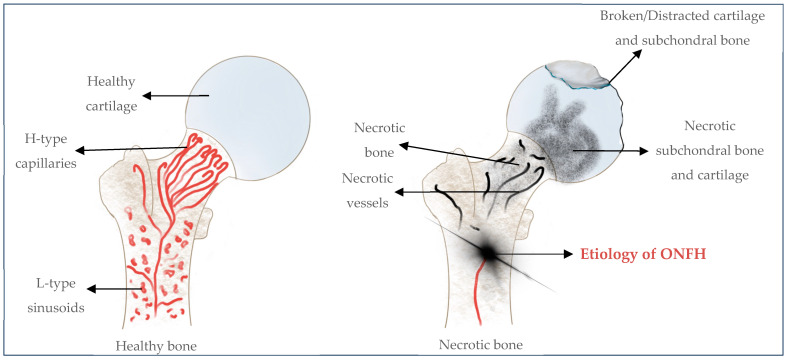
Difference between healthy and necrotic cartilage and bone shown in a femoral head drawing. ONFH, osteonecrosis of the femoral head.

**Figure 10 gels-10-00544-f010:**
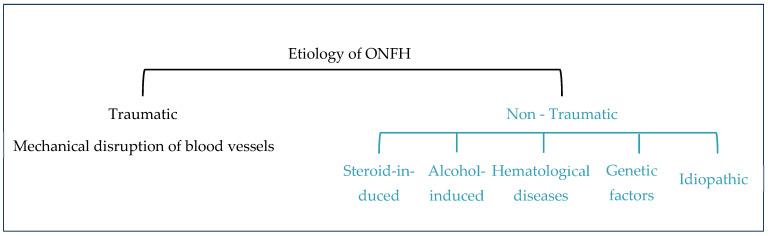
Scheme summarizing the etiology of ONFH. ONFH, osteonecrosis of the femoral head.

**Figure 11 gels-10-00544-f011:**
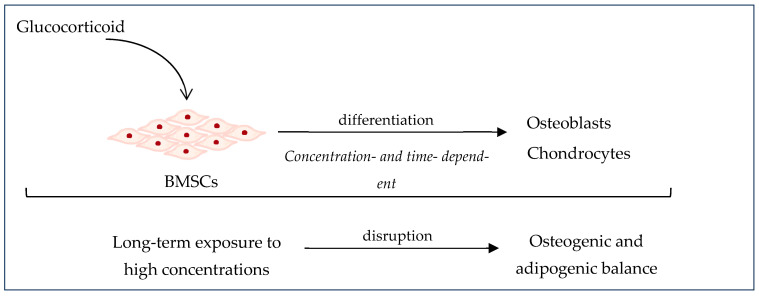
How glucocorticoids affect BMSCs. Glucocorticoids are known to affect the differentiation of bone marrow stem/stromal cells (BMSCs) into osteoblasts and cartilage cells, also known as chondrocytes. However, this differentiation effect is concentration- and time-dependent. Long-term exposure to a high concentration of glucocorticoids is known to favor adipogenic differentiation over osteogenic differentiation and to disrupt the osteogenic and adipogenic balance.

**Figure 12 gels-10-00544-f012:**
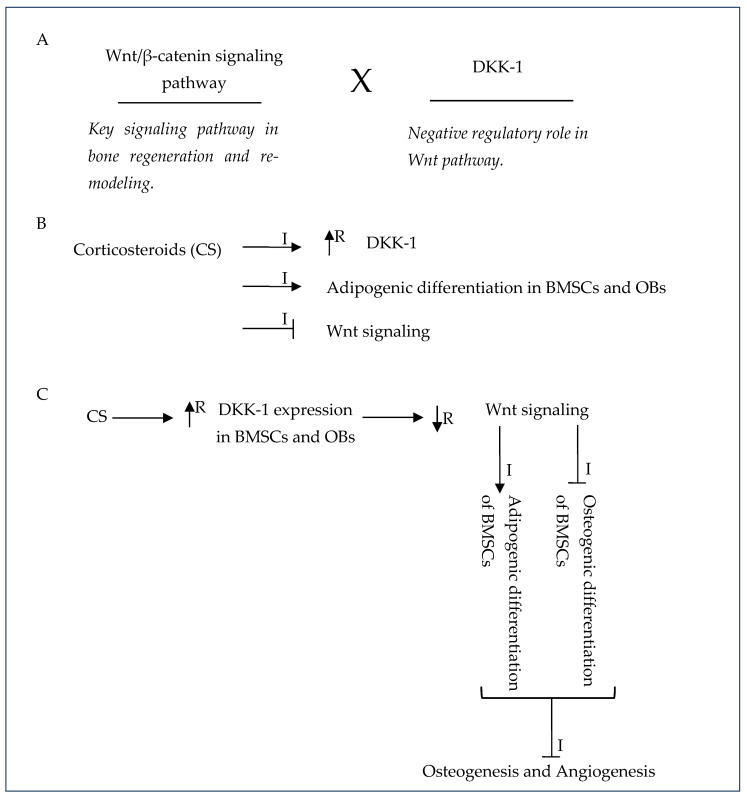
Main signaling scheme in steroid-induced ONFH. (A) The main signaling pathway in bone regeneration and remodeling is the Wnt/β-catenin pathway. Dickkopf-related protein 1 (DKK-1) has a negative regulatory role in the Wnt signaling pathway. (B.) Corticosteroids (CS) upregulate DKK-1 and adipogenic differentiation in bone marrow stem/stromal cells (BMSCs) and osteoblasts (OBs) while inhibiting the Wnt signaling pathway. (C) The detailed signaling pathway for Figure 12B. The long-term use of a high concentration of corticosteroids upregulates the expression of DKK-1 in BMSCs and OBs causing the downregulation of Wnt signaling. Downregulation of Wnt signaling causes the inhibition of osteogenic differentiation of BMSCs that indirectly negatively affects angiogenesis and leads to the upregulation of the adipogenic differentiation of BMSCs. ONFH, osteonecrosis of the femoral head.

**Figure 13 gels-10-00544-f013:**
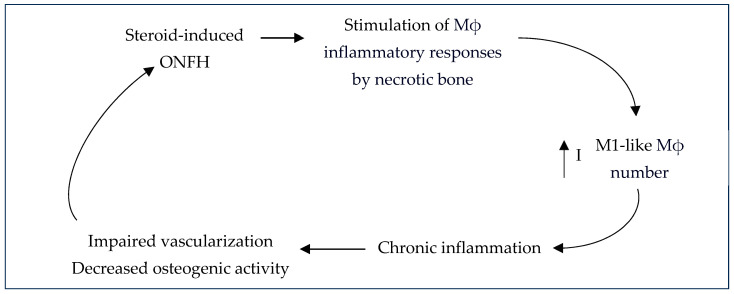
The steroid-induced ONFH cycle. Necrotic lesions or bone stimulate inflammatory responses by macrophages (Mϕ), which increases the number of M1-like Mϕ. M1-like Mϕ are known to promote chronic inflammation. This increased inflammation causes impaired vascularization and decreased osteogenesis and inhibits the bone regeneration process. This causes the aggressive progression of osteonecrosis of the femoral head (ONHF).

**Figure 14 gels-10-00544-f014:**
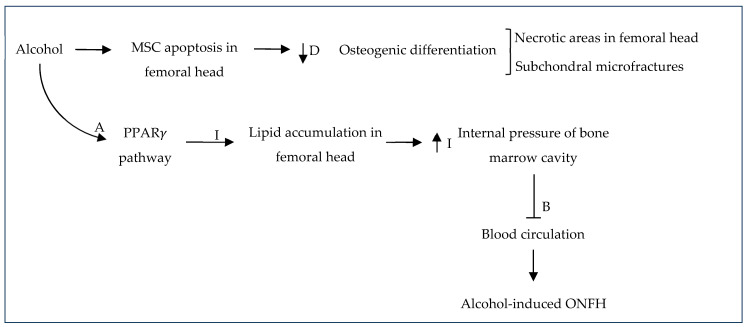
Alcohol-induced ONFH. Regular alcohol use causes apoptosis of the femoral head mesenchymal stem/stromal cells (MSCs) that decreases the osteogenic differentiation of stem cells. This decrease causes subchondral fractures and necrotic lesions in the femoral head. A regular alcohol intake is also known to activate the peroxisome proliferator-activated receptor gamma (PPAR*γ*) pathway, which is the key regulatory pathway in adipogenesis. The activation of this pathway induces lipid accumulation in the femoral head. Accumulated lipids cause an increase in pressure in the bone marrow cavity that suppresses the vasculature and blocks or impairs the blood circulation. This inadequate blood circulation causes osteonecrosis of the femoral head (ONFH).

**Figure 15 gels-10-00544-f015:**
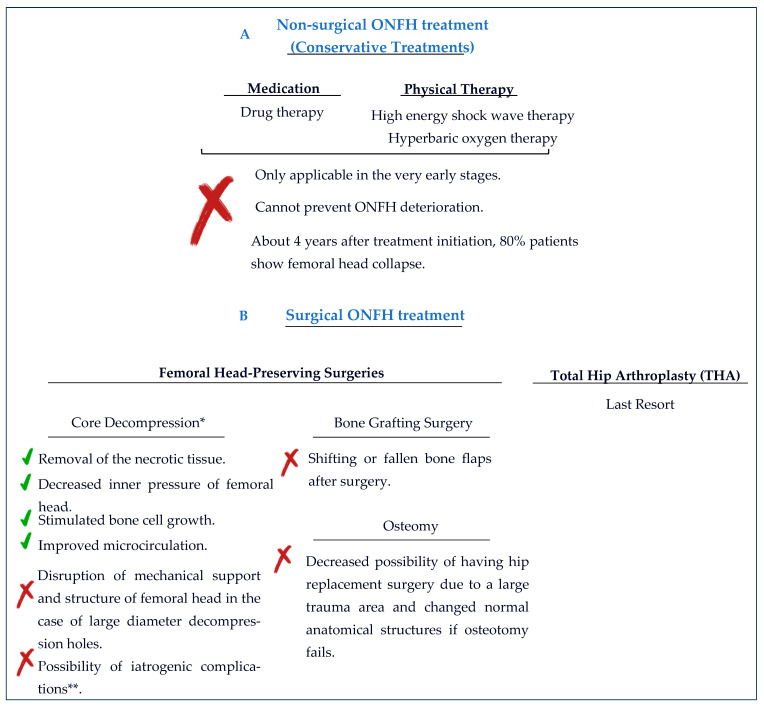
Treatment strategies for ONFH in the clinic. (**A**) Non-surgical, also known as conservative treatments. This treatment strategy is applicable only in the very early stages of osteonecrosis of the femoral head (ONFH) and can be divided into two: medication and physical therapies. However, unfortunately, 4 years after treatment almost 80% of patients show femoral head collapse. (**B**) Surgical ONFH treatment. This treatment strategy can be mainly divided into two: (1) femoral head-preserving surgeries that are mainly core decompression (CD), bone grafting surgery, and osteotomy; and (2) total hip arthroplasty. * In CD surgery, common operations to create intraosseous tunnels to stimulate the revascularization and repair of the femoral head are either (a) single tunnel decompression or (b) multiple epiphyseal drillings. ** Iatrogenic complications: subtrochanteric fracture or femoral head collapse.

**Figure 16 gels-10-00544-f016:**
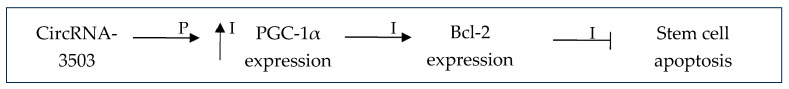
The signal transduction pathway of how CircRNA-3503 inhibits stem cell apoptosis. circRNA, circular RNA; PGC-1α, peroxisome proliferator-activated receptor-γ coactivator 1α; Bcl-2, B-cell leukemia/lymphoma protein-2.

**Figure 17 gels-10-00544-f017:**
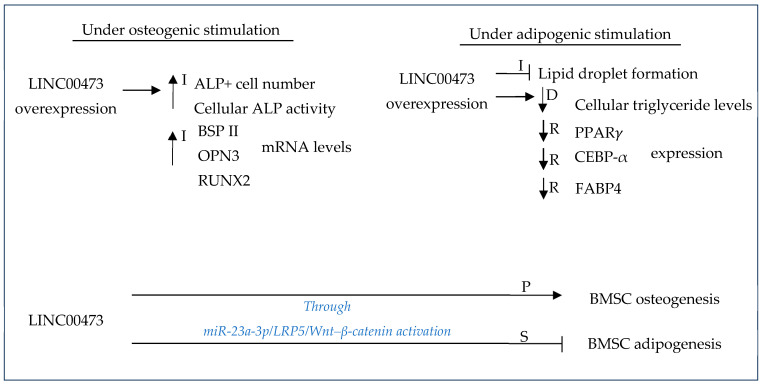
Effect of LINC00473 on BMSC osteogenesis and adipogenesis. Under osteogenic stimulation, the overexpression of long intergenic non-protein coding RNA 473 (LINC00473) is shown to increase the mRNA levels of the osteogenic markers bone sialoprotein II (BSPII), osteopontin 3 (OPN3), and Runt-related transcription factor-2 (RUNX2) while under adipogenic stimulation, LINC00473 reduced the expression of adipogenic markers peroxisome proliferator-activated receptor gamma (PPAR*γ*), CCAAT-enhancer-binding protein alpha (CEBP-α), and fatty acid-binding protein 4 (FABP4). As a result, LINC00473 promotes BMSC osteogenesis and suppresses BMSC adipogenesis through activation of the micro-RNA 23-a-3p/low-density lipoprotein receptor-related protein 5/Wnt–β-catenin (miR-23a-3p/LRP5/Wnt–β-catenin) pathway. ALP, alkaline phosphatase; BMSC, bone marrow stem/stromal cells.

**Table 1 gels-10-00544-t001:** FDA approval status of some synthetic polymers.

Synthetic Material Name	FDA Approval Status	Reference
Polycaprolactone (PCL)	Approved to be used clinically	[62]
Polylactic acid (PLA)	Approved for multiple applications clinically	[63]
Polyethylene glycol (PEG) *	Not approved in children, PEG 3350 limited use in adults	FDA website
Polyvinyl alcohol (PVA)	Approved to be used clinically	[64]
Poly-lactic-co-glycolic acid (PLGA)	Only 19 different products are approved to be used clinically	[65]

* Polyethylene oxide (PEO) is ethylene oxide with a molecular weight of more than 20,000 g/mol. Polyethylene glycol (PEG) is ethylene oxide with a molecular weight lower than 20,000 g/mol. Thus, the information is given as PEG, and PEO is not mentioned here. FDA, US Food and Drug Administration.

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
