# Peer review of "Hydrogel Use in Osteonecrosis of the Femoral Head"

_gels, 2024, doi:10.3390/gels10080544_

Round 1

Reviewer 1 Report

Comments and Suggestions for Authors

Dear Authors,

Thank you very much for submitting your work to Gels.

1.What is the current use of hydrogel worldwide?

2.What is the main finding for this research?

3.What is the true novelty compared with the previous publication?

4.The topic may be too broad and it seems too descriptive.

5.What do you want to emphasize for this research?

6.Could you provide the limitations for this study?

7.Woudl it be possible to make a true comparison, possibly using statistical analysis?

8.Please provide the clinical relevance.

Thank you very much.

Author Response

Dear Reviewer,

Here are our answers to your comments and questions.

Comment 1: What is the current use of hydrogel worldwide?

Response 1: Thank you very much. Having similar structure to extracellular matrix, hydrogels can be used in many applications from the perspective of tissue regeneration from skin regeneration to vascular and bone regeneration [Page 1, Abtsract, Lines 12-17]. Some of them are even used as commercial products such as skin regeneration gels that are used mostly after surgical operations and the use of them as contact lenses is very common – and understandable when we think of the first biological hydrogel product was a contact lens in 1960s.

Comment 2: What is the main finding for this research?

Response 2: Thank you very much for your question however this paper is not a research paper but a review paper therefore we can not mention about any research findings. Yet we can still emphasize that there is no previous review article that summarizes the use of hydrogels in the femoral head necrosis[Page 45, Paragraph no 2,  Lines 108-110, we added in order to emphasize : “In the literature until now, there is no review paper focusing on the hydrogel use specifically on ONFH research and treatment. Thus, as the first review paper focusing on this subject, we hope this review would be inspiring for the ONFH research.]. This is the main focus of this paper.

Comment 3: What is the true novelty compared with the previous publication?

Response 3: Thank you for your question. The true novelty of this publication is its being the first publication summarizing the use of hydrogels in clinics or basic research specifically in the area of osteonecrosis of the femoral head. There is no previous review paper focusing on hydrogel use in the treatment of osteonecrosis of the femoral head [Page1, Abstract, Lines 20-22].   [Page 45, Lines 108-110, we added in order to emphasize : “In the literature until now, there is no review paper focusing on the hydrogel use specifically on ONFH research and treatment. Thus, as the first review paper focusing on this subject, we hope this review would be inspiring for the ONFH research.]. As a researcher in the area of orthopaedics and vascularization and focusing on the osteonecrosis of the femoral head, the first author of this paper can easily say that on the literature it is hard to find a neat information solely handling the femoral head osteonecrosis research. Thus as the authors of this paper we aimed to write a review which would be a milestone for the researchers who would like to study hydrogels and their application in femoral head osteonecrosis. Additionally, since the use of hyrogels in basic research or clinical trials is a new area and getting popular recently, we believe that having a paper summarizing the developments and the research in this area, will contribute to acceleration of the basic studies and clinical trials.

Comment 4: The topic may be too broad and it seems too descriptive.

Response 4: Thank you very much for your comment. The osteonecrosis of the femoral head is a disability occurs solely on the specific area of the femoral bone : proximal femor. This actually a special area of the femoral bone that has a prime importance for also hip health. Thus we do not think the topic is too broad. But in order to focus on this specific part of the bone, we had to handle the topic from the perspective of  the vascularization and the bone  regeneration because osteonecrosis mostly occurs because of the insufficient supply of blood to the regarding bone area. Thus we think that understanding this disability needs not only the knowledge about bone but also about the vascularization. For the materials section, we also had to give brief information about the other biomaterials rather than composite grafts to provide the readers an easy understanding of the importance of hydrogel use in femoral head necrosis and what makes hydrogels better than other grafting.

Comment 5: What do you want to emphasize for this research?

Response 5: Thank you very much for your question. Although this is not a research article, with this review article we aimed to to write a review which would be a milestone for the researchers who would like to study hydrogels and their applications in femoral head osteonecrosis. Thus as we mentioned in the different parts of the paper [Page 9, Paragraph no 1, Lines 350-358; Page 10, Paragraph no 1, Lines 380-391; Page 15, Paragraphi no 2, Lines 392-400; Page 17, Paragraph no 5,  Lines 71-715; Page 18, Paragraph no 1, Lines 716-718], we emphasized the importance of hydrogel use and its use in the treatment of bone and vascular regeneration.  Additionally, we also emphasized about the importance of osteogenic-angiogenic coupling in the treatment of osteonecrosis of the femoral head as a perspective to the researchers of this study area [Page 44, Paragraph no 8, Lines 100-104; Page 45, Paragraph no 1, Lines 105-107]. Additionally, since the use of hyrogels in basic research or clinical trials is a new area and getting popular recently, we believe that having a paper summarizing the developments and the research in this area, will contribute to acceleration of the basic studies and clinical trials.

Comment 6: Could you provide the limitations for this study?

Response 6: Thank you very much for your question. The limitaion of this study can be the limited basic and clinical studies regarding the hyrogel use in osteonecrosis of the femoral head. The use of hydrogel in this area is almost very recent and the lack of small animal models is one of the problem that impedes the accelerated research in this area.

Comment 7: Would it be possible to make a true comparison, possibly using statistical analysis?

Response 7: Thank you very much for your question. The authors are not sure if we understood the question well, yet this paper as a review paper with limited number of previous research articles with many different growth factors or ion combinations mixed with different hydrogel materials, we do not think any statistical comparision is possible based on the number research articles and variables.

Comment 8: Please provide the clinical relevance.

Response 8: Thank you very much for your comment. This is not a research paper therefore we believe we cannot mention about any findings of ours on the clinical relevance. In our paper we already mentioned about the clinical trials of hydrogels in femoral head necrosis however what we can say is, since the use of hyrogels in basic research or clinical trials is a new area and getting popular recently, we believe that having a paper summarizing the developments and the research in this area will contribute to acceleration of the basic studies and clinical trials. The clinical relevance is given on  [Page 44, Paragraph no 3-5, Lines 70-85] and also on [Page 44,  Paragraph no 7, Lines 97-99].

Reviewer 2 Report

Comments and Suggestions for Authors

The authors have attempted to provide a comprehensive overview of ONFH, however the information throughout the manuscript is not well focused. For example, they have presented a lot about bone grafts and BTE in general in the introduction section without making a logical connection to ONFH. The same defocusing happened with hydrogels, many materials are listed in part 3 without direct reference to ONFH. 

For better understanding, especially for those readers who are not familiar with the ONFH, it would be helpful to differentiate the materials according to their clinical applicability or according to their research and development, i.e. whether the materials discussed are already approved for clinical treatment or whether some of them are still under development due to various challenges, which is an indication of future potential. It is also important to clearly identify the challenges faced during ONFH development, be they technical issues, scientific understanding or regulatory barriers?

As this is a comprehensive review that summarizes the current state of the field and identifies future trends, it is necessary to clearly state in the manuscript the criteria for the selection of literature and the time period in which the publications appeared. 

Comments on the Quality of English Language

Writing style and sentence organization should be more scientific and clear.

Author Response

Dear Reviewer,

Thank you very much for your time to assess our paper.

Here are our answers to your comments and questions.

Comment 1: The authors have attempted to provide a comprehensive overview of ONFH, however the information throughout the manuscript is not well focused. For example, they have presented a lot about bone grafts and BTE in general in the introduction section without making a logical connection to ONFH. The same defocusing happened with hydrogels, many materials are listed in part 3 without direct reference to ONFH. 

Response 1: Thank you very much for your comment. In introduction, we actually mentioned about ONFH mostly [Page 1, Paragraph no 2, Lines 40-46; Page 2, Paragraph no 1, Lines 47-53]. We also think that the connection between hydrogel grafts and ONFH is given clearly in the introduction paragraph after mentioning about the superiority of hydrogels in vascularization and importance of vascularisation in bone regeneration [Page 1, Paragraph no 1, Lines 37-39].

I think the reviewer kindly mentioned about not the “1. Introduction” part but the “2. Bone and Bone Grafting” part. To be honest, we wanted to mention about bone and graft materials a bit detailed because for any researcher without at least any introductory knowledge of bone and bone grafts, understanding ONFH may be a bit complex because it also is a vascular disease. Because of this we also mentioned about the importance of osteogenic-angiogenic coupling in bone remodeling [Page 3, Paragraph no 1, Lines 111-122; Page 4, Paragraph 1, Lines 123-127] to prepare the reader to the subsection “Osteonecrosis of the Femoral Head”- which is also an introductory section to prepare the reader to a more detailed section as “5. Osteonecrosis of the Femoral Head” [Page 22].  

We also think it is important to have a basic knowledge about bone graft types, their pros and cons and the desired properties of graft materials,  for readers to be able to locate the hyrogels properly in their minds and to keep the finger on the pulse which property of a graft is important and why it is important.

The other point the reviewer kindly mentioned is the defocusing of the materials listed in part without direct reference to ONFH. As authors we understand the concern of the reviewer but, we kindly have to mention here that hydrogel use in osteonecrosis of the femoral head is a very new area of study and there are no certain materials that are set to be used in ONFH treatment with hydrogels. Therefore, we decided to choose and to focus on the materals that are mostly used in bone regeneration area and hyrogels together, througout which of them have been used in hydrogel and ONFH research already and which of them are promising.

For figure please check the word file 

We also added Figure 3 [Page 7] in order to focus the reader more to ONFH for the next sections and to support the introductory ONFH information in the section 2.

Figure 3. Brief summary of ONFH. ONFH is commonly seen through young adults of age 30s or 40s and it generally occurs bilaterally in almost %50 of the clinical cases. It may be traumatic or non-traumatic. Although the main cause behind ONFH is not elucidated yet, hypercoagulation. Elevated bone marrow pressure of the proximal femur, vascular dysfunction and bone marrow fat embolism are known as the contributing factors. The necrotic debris in ONFH releases the pro-inflammatory factors which leads to persistent chronic inflammation that ends up with increased bone resorption, disruption of the balance between bone anabolism and bone catabolism, and decreased bone formation concomitantly. It further would lead to subchondral fractures and secondary arthritis, accompanying hip pain, gait disorders, limited motion which shows the progression of the disease to femoral hip collapse which would lead to physical disability.

Comment 2: For better understanding, especially for those readers who are not familiar with the ONFH, it would be helpful to differentiate the materials according to their clinical applicability or according to their research and development, i.e. whether the materials discussed are already approved for clinical treatment or whether some of them are still under development due to various challenges, which is an indication of future potential. It is also important to clearly identify the challenges faced during ONFH development, be they technical issues, scientific understanding or regulatory barriers?

Response 2: Thank you very much for your idea. We totally agree that differentiating the materilas depending on their application in clinics or research would be a very good point for readers to understand yet, as the reviewer would also be aware not all materials are applicable in every disease situation. For example BMP-2 is approved by FDA but only for use “in anterior lumbar spinal fusion and tibial non union fractures” [Page 15, Paragraph no 4, Lines 598-599] and BMP-7 is approved by FDA only for use in “posterolateral spinal lumbar fusion and for complicated permanent tibial pseudoarthritis” [Page 15, Paragraph no 4, Lines 599 – 601]. If the use of the synthetic material is totally approved by FDA, we generally tried to mentioned it in the sub-sections related to them as we did for PEG [Page 13] and PCL [Page 13] however we totally agree with the reviewer about mentioning the other synthetic materials’ approval status thus we searched them on FDA’s web site and noted them down as such: [Page 13].

Synthetic Material Name

FDA Approval Status

Reference

Polycaprolactone (PCL)

Approved to be used clinically

115

Polylactic Acid (PLA)

Approved for multiple applications clinically

117

Polyethylene Glycol (PEG) *

 Not approved in children, PEG 3350 limited use in adults

FDA website

Polyvinyl Alcohol (PVA)

Approved to be used clinically

116

Poly-lactic-co-glycolic acid (PLGA)

Only 19 different products are approved to be used clinically

118

Table 1. FDA Approval status of some synthetic polymers

*Polyethylene oxide PEO is ethylene oxide with a molecular weight more than 20,000 g/mol and polyethylene glycol is ethylene oxide with a molecular weight lower than 20,000 g/mol. Thus the information is given as PEG and PEO is not mentioned here.

The reviewer also suggested “It is also important to clearly identify the challenges faced during ONFH development, be they technical issues, scientific understanding or regulatory barriers?”

We agree with the reviewer. However we have already mentioned about the problems related to ONFH in the part “5.1. Pathology of the Femoral Head Necrosis” by emphasizing why it is hard to characterize the very early pathological changes in ONFH and also mentioned about the clinical treatments of the ONFH under the same sub-section with pros and cons and also emphasized that there is no standard surgical treatment defined worldwide [Page 30, Paragraph no. 1-3, Lines 1241-1271]

In detail, actually the problem with ONFH as we mentioned in the first paragraph of the section “5.1. Pathology of the Femoral Head Necrosis” it is hard to characterize the very eraly pathological changes of ONFH which does not let us to understand the molecular mechanism behind ONFH clearly.

However, because we really attach importance to the reviewer’s suggestion, we also added a figure that can summarize the clinical treatment strategies of ONFH with pros and cons [Figure 15, Page 29].

For figure please check the word file 

Figure 15. Treatment strategies for ONFH in clinic. A. Non-surgical aka conservative treatments. This treatment strategy is applicable only in the very early stages of ONFH and can be divided into two as medication and physical therapy however unfortunately 4 years after the treatment almost 80% of the patients further transform into femoral head collapse. B. Surgical ONFH Treatment. This treatment strategy can be mainly divided into two as: (1) Femoral head preserving surgeries which are mainly core decompression, bone grafting surgery and osteotomy and (2) Total hip arthroplasty. * In core decompression surgery the common operations to create intraosseous tunnels to stimulate the revascularization and repair of femoral head are either (a) single tunnel decompression or (b) multiple epiphyseal drillings. ** Iatrogenic complications: subtrochanteric fracture or femoral head collapse.

Comment 3: As this is a comprehensive review that summarizes the current state of the field and identifies future trends, it is necessary to clearly state in the manuscript the criteria for the selection of literature and the time period in which the publications appeared. 

Response 3. Thank you very much for the reccommendation. We totally are agree. Thus we added “The literature research was done via PubMed and Google Scholar by using the keywords as “hydrogel + osteonecrosis of the femoral head”, “hydrogel + femoral head necrosis” and all accessible studies from any time interval until 2023 related to hydrogel use in osteonecrosis of the femoral head was included in this paper [Page 2, Paragraph no 2, Lines 62-65].

Reviewer 3 Report

Comments and Suggestions for Authors

The manuscript "Hydrogel Use in Osteonecrosis of the Femoral Head" by Bal Zeynep and Takakura Nobuyuki provides a comprehensive review of hydrogels' role in treating ONFH osteonecrosis. This topic is highly relevant given the complexities associated with ONFH and the potential benefits of hydrogel applications in regenerative medicine. However, the manuscript requires major revisions before it can be considered for publication. Below, I outline specific comments, questions, and suggestions to improve the manuscript's quality:

(1) The manuscript provides a broad overview of bone and vascular biology, hydrogel properties, and their applications. However, some sections lack clear focus and depth, especially regarding the specific mechanisms by which hydrogels contribute to the treatment of ONFH.

(2) The structure could be improved by integrating the sections more coherently. For example, the transition between general bone biology and the specific application of hydrogels in ONFH could be more seamless.

(3) The abstract should be more concise and clearly state the main findings and conclusions of the review. Consider including specific examples of hydrogel applications in ONFH.

(4) The introduction provides a good overview but could be more focused on the specific challenges and current treatments of ONFH, setting the stage for the discussion on hydrogels. Can you elaborate on the limitations of current ONFH treatments and how hydrogels address these limitations?

(5) The bone and bone grafting section is comprehensive but somewhat tangential. It could be shortened to focus more on aspects directly relevant to hydrogel use in ONFH. How do the properties of hydrogels compare with traditional bone graft materials in terms of efficacy and safety?

(6) On page 5, the authors should consider discussing the incorporation of octacalcium phosphate (OCP) into hydrogels for bone tissue engineering applications. OCP has shown promising results in enhancing osteoconductivity and promoting bone regeneration, making it a valuable addition to hydrogel formulations. Incorporating OCP could improve the efficacy of hydrogels in treating bone defects. Relevant literature on this topic includes studies demonstrating the benefits of OCP in bone tissue engineering, such as "https://doi.org/10.3390/ijms241713135" and "https://doi.org/10.1016/j.mtcomm.2022.103312".

(7) While the properties of hydrogels are well-described, more emphasis should be placed on their specific interactions with bone tissue and their role in promoting vascularization. For example, the authors should include more detailed examples of how hydrogels promote osteogenesis and angiogenesis in ONFH.

(8) What are the most promising clinical studies on hydrogel applications in ONFH, and what do their results indicate?

(9) The manuscript could benefit from a deeper mechanistic understanding of how hydrogels interact with bone and vascular tissues at the molecular level. For example, what are the key molecular pathways involved in the interaction between hydrogels and bone tissue, and how do they contribute to bone regeneration?

(10) It would be useful to perform a comparison of various types of hydrogels and their specific advantages and disadvantages by evaluating their biocompatibility, degradability, and mechanical strength.

Author Response

Dear Reviewer,

Thank you very much for your time to assess our paper and sharing your detailed and important ideas with us.

Here are our answers to your comments and questions.

We kindly suggest you to check the additional word file for the figures and answers.

The manuscript "Hydrogel Use in Osteonecrosis of the Femoral Head" by Bal Zeynep and Takakura Nobuyuki provides a comprehensive review of hydrogels' role in treating ONFH osteonecrosis. This topic is highly relevant given the complexities associated with ONFH and the potential benefits of hydrogel applications in regenerative medicine. However, the manuscript requires major revisions before it can be considered for publication. Below, I outline specific comments, questions, and suggestions to improve the manuscript's quality:

Comment 1: The manuscript provides a broad overview of bone and vascular biology, hydrogel properties, and their applications. However, some sections lack clear focus and depth, especially regarding the specific mechanisms by which hydrogels contribute to the treatment of ONFH.

Response 1: Thank you very much for your comment. Unfortunately hydrogel use in ONFH basic research or clinics is a very new area thus we cannot detailly mention about the specific signal pathways regarding the specific hydrogels. However we do agree that the lack of signal transduction information in our paper so, we decided to compansate this deficiency by adding defined signaling pathways or basic schemes in some sections of our paper such as: (please check the word file)

Comment 2: The structure could be improved by integrating the sections more coherently. For example, the transition between general bone biology and the specific application of hydrogels in ONFH could be more seamless.

Response 2: On behalf of the authors, as the first author I worked on structure a bit more. Thank you very much.

Comment 3.  The abstract should be more concise and clearly state the main findings and conclusions of the review. Consider including specific examples of hydrogel applications in ONFH.

Response 3. Thank you very much for your suggestion. We would like to emphasize at least the clinical studies in the abstract but the word limit for abstract is 200 characters. However, we added a short information as “..[4] ONFH and hydrogel use in ONFH with the literature studies which show promising results in limited clinical studies” to mention at least the promising future of hydrogel use in ONFH [Page 1, Paragraph no 1, Lines 19-20].

Comment 4: The introduction provides a good overview but could be more focused on the specific challenges and current treatments of ONFH, setting the stage for the discussion on hydrogels. Can you elaborate on the limitations of current ONFH treatments and how hydrogels address these limitations?

Response 4: Thank you very much for your valuable suggestion. Actually we mentioned about the specific challenges and current treatments of ONFH in the part “5.1. Pathology of the Femoral Head Necrosis”[Page 27, Paragraph no 1, Lines 1131-1133, Page 28, Paragraph no 1, Lines 1134-1137, Page 30, Paragraph 1-3, Lines 1241-1271]. We did not want to mention these in detail in the introductory sections for two reasons: (1) in order to refrain from the repetitions because we think that if we mention it in the subtitle related to ONFH in section 2, the reader may get bored or may be prejudisced that s/he will read the same information in the Section 5, Subsecton 5.1 and may omit this section ; (2) to only focus on the general information about ONFH to prepare the reader to a very soft and easy pass to hydrogel treatment in ONFH.

On behalf of the authors, as the first author, I, to be honest cannot conclude that if hydrogels can adress the whole limitations yet because we still do not have a defined surgery standard method in treatment of ONFH, additionally the molecular mechanisms behind ONFH are not clarified yet also [and actually I did not want to mention this in the paper on purpose because we have an ongoing project related to this topic and as the first author I find it ethically not proper to indirectly promote us in a review article]. Thus hydrogels are one of the grafts that are recently started to be used in ONFH studies and they are just promising  as other materials. However why they are promising is mentioned on [Page 30, Paragraph no 4, Lines 1272-1279].

However, based on the valuable opinion of the reviewer, we also added Figre 15 [Page 29] in order to summarize the treatment strategies. Please check the word file.

Comment 5: The bone and bone grafting section is comprehensive but somewhat tangential. It could be shortened to focus more on aspects directly relevant to hydrogel use in ONFH. How do the properties of hydrogels compare with traditional bone graft materials in terms of efficacy and safety?

Response 5: Thank you very much for your comment and recommendation. We added “There are also granular of sponge bone grafts that can be used after core decompression surgery however these grafts compacted into the drills are reported to show osteoinduction only in the region of the drilled tunnels [32]. Thus, hydrogels with their ECM-like properties..” as a comparision of other materials used after core decompression surgery with hydrogels [Page 8, Paragraph no. 2, Lines 311-314].

However if this paper would be a review paper related to bone and hydrogel use in bone regeneration or osteonecrosis I think we would have a chance to focus directly on the aspects directly relevant to hydrogel use in ONFH and would be able to compare the grafts yet, the studies regarding hyrogel use in ONFH are limited for now and we think if we can only focus on those materials this paper would just be a summary of the studies done until now. What we want is to broaden the perspective of the researchers and clinicans and to show them different point of views and to mention them that there are other materials which can be used as a candidate also.

Comment 6: On page 5, the authors should consider discussing the incorporation of octacalcium phosphate (OCP) into hydrogels for bone tissue engineering applications. OCP has shown promising results in enhancing osteoconductivity and promoting bone regeneration, making it a valuable addition to hydrogel formulations. Incorporating OCP could improve the efficacy of hydrogels in treating bone defects. Relevant literature on this topic includes studies demonstrating the benefits of OCP in bone tissue engineering, such as "https://doi.org/10.3390/ijms241713135" and "https://doi.org/10.1016/j.mtcomm.2022.103312".

Response 6: Thank you very much for your suggestion. We decided to mention about OCP on page 14, under the sub-section of “3.6. Calcium Phosphates” as “Recently also bioactive octocalcium phosphates (OCP) are becoming popular in bone regeneration with hydrogels area because of their similarity to hydroxyapatite and ability to promote charge transfer between the material and the tissue, and they actually can be converted to hydroxyapatites in body because they are the precursors of hydroxyapatite” [Page 15, Paragraph no 2, Lines 577-583 ] and the citations are added to reference list as 71 to 74 (highlighted with yellow).

Comment 7:  While the properties of hydrogels are well-described, more emphasis should be placed on their specific interactions with bone tissue and their role in promoting vascularization. For example, the authors should include more detailed examples of how hydrogels promote osteogenesis and angiogenesis in ONFH.

Response 7: Thank you for your suggestion. We think you are right however not many signal pathways are defined in related papers in which hyrogels in ONFH are studies. Additionally we already summarized the papers in general under the section “5.2. Hydrogel Use in ONFH in the Literature” [Page 30-44]. We think if we also give every detail in the paper about the materials that are studied in detail it would not be ethical. However in order to compansate this deficiency in our paper, we added some important signaling pathways or basic schemes to our paper as given in Response 1.

Comment 8: What are the most promising clinical studies on hydrogel applications in ONFH, and what do their results indicate?

Response 8: Thank you very much for your question. Unfortunately there are not many clinical studies on Hydrogel use in ONFH treatment yet, the main two clinical studies were summarized and criticsed in our paper under the sub-section “5.2.2. Clinical Studies” [Page 44, Paragraph no 3-5, Lines 70-85]. If they are needed to be summarized more the single shot injection of gelatin hydrogel + rhFGF2 treatment seems promising in human samples because this treatment is shown to increase the joint preservation rate more than 65%. However they did not checked the vascularisation and did not focused on the effects of the hydrogel growth factor combination’s effect on vascular changes.

Comment 9:  The manuscript could benefit from a deeper mechanistic understanding of how hydrogels interact with bone and vascular tissues at the molecular level. For example, what are the key molecular pathways involved in the interaction between hydrogels and bone tissue, and how do they contribute to bone regeneration?

Response 9: Thank you very much for your comment and suggesstion. My apologies on the behalf of all authors, however I should again here give a similar answer with the previoues comment no 7 that we are not eligible to summarize the signal pathways from the mechanistic perspective specifically on hydrogel use in ONFH because the hydrogel use in ONFH is a very new study area and even the molecular back ground of ONFH itself is not very well clarified yet.

Under these circumstances we can only speculate from the general informations like the effect of shear stress on vascularization or hydrogel stiffness effects on stem cell differentiation and so on. However none of them would be direct information about the circumstances under ONFH. This is why we kindly do not want to mention these in our paper in order not to direct researcher in possible wrong direction. 

Comment 10:  It would be useful to perform a comparison of various types of hydrogels and their specific advantages and disadvantages by evaluating their biocompatibility, degradability, and mechanical strength.

Response 10: Thank you very much for your valuable comment and suggesstion. It would be very well to do that however unfortunately we focused on the hydrogels used in ONFH treatment in this paper. Thus we think,  either the samples size or the study results are not proper for such a comparision.

Round 2

Reviewer 1 Report

Comments and Suggestions for Authors

Dear Authors,

Thank you very much preparing revised manuscript.

Reviewer 2 Report

Comments and Suggestions for Authors

All issues and comments have been properly addressed in the revised version

Reviewer 3 Report

Comments and Suggestions for Authors

Accept in the present form.